# SOTAlign: Semi-Supervised Alignment of Unimodal Vision and Language Models via Optimal Transport

**Simon Roschmann** [* 1 2 3 4]  **Paul Krzakala** [* 5 6]  **Sonia Mazelet** [6]  **Quentin Bouniot** [1 2 3 4 5]  **Zeynep Akata** [1 2 3 4]

## Abstract

The Platonic Representation Hypothesis posits that neural networks trained on different modalities converge toward a shared statistical model of the world. Recent work exploits this convergence by aligning frozen pretrained vision and language models with lightweight alignment layers, but typically relies on contrastive losses and millions of paired samples. In this work, we ask whether meaningful alignment can be achieved with substantially less supervision. We introduce a semi-supervised setting in which pretrained unimodal encoders are aligned using a small number of image–text pairs together with large amounts of unpaired data. To address this challenge, we propose SOTAlign, a two-stage framework that first recovers a coarse shared geometry from limited paired data using a linear teacher, and then refines the alignment on unpaired samples via an optimal-transport-based divergence that transfers relational structure without overconstraining the target space. SOTAlign effectively leverages unpaired images and text, learning robust joint embeddings across datasets and encoder pairs, and significantly outperforming supervised and semi-supervised baselines. Code is available at https://github.com/ExplainableML/SOTAlign.

## 1. Introduction

Vision-language models (VLMs) learn a shared embedding space for images and text, enabling zero-shot transfer to unseen concepts and domains. Since CLIP (Radford et al., 2021) and ALIGN (Jia et al., 2021), the dominant paradigm has relied on large-scale contrastive training on

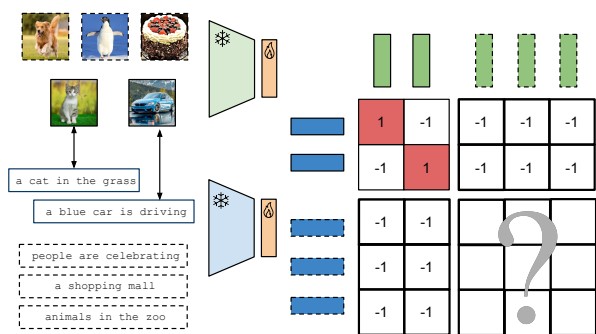

*Figure 1. Semi-supervised vision-language alignment.* We tackle the alignment of frozen unimodal encoders where paired data (red blocks) are scarce but unpaired data are abundant. The key challenge is: how to define a training signal for unpaired data when ground-truth cross-modal correspondences are missing?

paired image-text data, with performance improving predictably as supervision scales. While effective, this approach requires hundreds of millions of paired samples (Cherti et al., 2023), making VLMs costly to train and difficult to adapt when paired data are limited. This issue arises in many critical applications, such as specialized scientific, medical, or industrial domains, where collecting large-scale annotations is expensive, time-consuming, or infeasible.

In this paper, we investigate vision–language alignment beyond large-scale supervision, asking whether meaningful alignment can be achieved from pretrained encoders using only a small number of paired samples together with abundant unimodal data. We posit that, under the Platonic Representation Hypothesis (Huh et al., 2024), unimodal models should already encode compatible semantic structures, making such alignment possible with minimal supervision.

**SOTAlign Overview.** We focus on training lightweight alignment layers on top of pretrained unimodal encoders. In this setting, we first demonstrate that meaningful cross-modal alignment can be recovered from very *few paired samples* using simple linear methods, providing empirical support for the Platonic Representation Hypothesis. Then, we introduce SOTAlign (Semi-supervised Optimal Transport-based Alignment), a simple approach that enables to further refine such alignment by leveraging *large unimodal datasets*, achieving state-of-the-art results in this *semi-supervised*

---
[*]Equal contribution [1]Helmholtz Munich [2]Technical University of Munich [3]Munich Center for Machine Learning [4]Munich Data Science Institute [5]Télécom Paris [6]École Polytechnique. Correspondence to: Simon Roschmann <simon.roschmann@tum.de>.

*Proceedings of the 43rd International Conference on Machine Learning*, Seoul, South Korea. PMLR 306, 2026. Copyright 2026 by the author(s).

setting. SOTAlign relies on KLOT, an *optimal-transport-based divergence* that enables to transfer the initial geometric structure of the linear teacher while allowing sufficient flexibility to avoid underfitting. Critically, we derive the *explicit gradient* of KLOT divergence, removing the memory bottlenecks that have limited the scalability of previous optimal-transport-based alignment methods. In the experimental section, we implement strong supervised and semi-supervised baselines and demonstrate the superior performances of SOTAlign across a wide range of downstream tasks. We carefully explore the robustness of SOTAlign to a variety of factors such as the number of paired samples, the number of unimodal samples, and the pretrained unimodal models. Critically, we also show that SOTAlign can leverage samples from multiple sources simultaneously, for example combining unimodal images from ImageNet and captions from CC12M to improve performances on COCO despite a significant distribution shift.

In short, we make the following contributions:

- We show that meaningful vision–language alignment can be recovered from very few paired samples using simple linear methods.

- We introduce SOTAlign, a semi-supervised approach that leverages unpaired unimodal data to achieve state-of-the-art alignment.

- We propose KLOT, a novel optimal-transport-based divergence, and fully address the memory bottlenecks that plagued prior OT-based methods.

- We validate the robustness of SOTAlign through extensive experiments across tasks, datasets, and encoders.

## 2. Related Work

**Vision–Language Models.** VLMs learn a joint embedding space for images and text, enabling zero-shot transfer across downstream tasks. CLIP (Radford et al., 2021) established this paradigm through large-scale contrastive pretraining on 400 million image–text pairs. Subsequent work, including ALIGN (Jia et al., 2021), SigLIP (Zhai et al., 2023), and SigLIP 2 (Tschannen et al., 2025), focused on scaling data and refining contrastive objectives to improve performance. These efforts are consistent with empirical scaling laws observed for CLIP-style models (Cherti et al., 2023), but also highlight a central limitation: achieving state-of-the-art performance *requires millions or billions of paired samples*, which is impractical in many settings and modalities.

More recently, OT-CLIP (Shi et al., 2024) proposed an Optimal Transport (OT) interpretation of the InfoNCE objective (Oord et al., 2018), viewing contrastive learning as inverse OT with a fixed identity transport plan. We adopt this perspective in the present work, but extend it beyond fully supervised settings by allowing target transport plans that are not restricted to the identity. Moreover, we derive an explicit expression for the gradient of the resulting objective (Theorem 5.1), removing the memory bottlenecks that have limited OT-based approaches to small batch sizes.

**The Platonic Representation Hypothesis.** Huh et al. (2024) posit that neural networks trained on different modalities, architectures, or objectives tend to converge toward compatible latent representations that reflect shared underlying structure in the data. In the context of vision-language models, this perspective suggests that pretrained unimodal image and text encoders may already produce semantically aligned representations, even in the absence of explicit cross-modal training.

This observation motivates an alternative approach to VLM construction, in which the pretrained encoders are kept frozen and only lightweight alignment layers are learned to reconcile their representation spaces. Several recent works adopt this paradigm, demonstrating that strong vision–language performance can be achieved without training multimodal models from scratch (Vouitsis et al., 2024; Zhang et al., 2025a; Maniparambil et al., 2025; Huang et al., 2025). UOT-RCL (Han et al., 2025) additionally leveraged OT to make the alignment of frozen encoders robust to noisy labels. Our work follows this line of research, but focuses on regimes where paired supervision is severely limited.

**Low-Supervision Alignment.** A growing body of work has explored alignment under weak, limited, or absent supervision. In unimodal settings, Jha et al. (2025) show that text embeddings can be aligned across representation spaces without paired data. Extending this idea to cross-modal alignment, Maniparambil et al. (2024) and Schnaus et al. (2025) demonstrate that vision–language representations can also be matched without supervision, but rely on quadratic assignment problem solvers that scale only to a few hundred samples, limiting applicability.

Closer to our setting, S-CLIP (Mo et al., 2023) introduces a semi-supervised framework in which optimal transport defines target similarities between unpaired images and paired captions, with promising results for domain adaptation of CLIP. In contrast, we define target similarities even between unpaired images and unpaired captions, enabling effective use of large-scale unimodal data on both sides (Figure 1). SUE (Yacobi et al., 2025) also considers semi-supervised vision–language alignment, but is limited to a single dataset and a single downstream task. Our work generalizes this setting across tasks, datasets, and encoder combinations. Finally, STRUCTURE (Gröger et al., 2025) augments InfoNCE with a regularization term encouraging preservation of unimodal geometry. While evaluated in supervised settings, this idea could in principle leverage unpaired data and is therefore included as a baseline in our experiments.

# 3. Methodology

**Notations.** For $u, v \in \mathbb{R}^d$, we denote the cosine similarity by $k(u, v) = \frac{\langle u, v \rangle}{\|u\| \|v\|}$. We stack batches of $n$ vectors as matrices in $\mathbb{R}^{n \times d}$. Given $U \in \mathbb{R}^{n \times d}$ and $V \in \mathbb{R}^{m \times d}$, we define the affinity matrix $K[U, V] \in \mathbb{R}^{n \times m}$ with entries

$$K[U, V]_{i,j} = k(U_i, V_j).$$

We define the (row-wise) Softmax normalization as

$$\text{Softmax}_\varepsilon(K)_{i,j} = \frac{\exp(K_{i,j}/\varepsilon)}{\sum_{k=1}^n \exp(K_{i,k}/\varepsilon)}.$$

## 3.1. Problem Formulation

We denote $d_x$ (resp. $d_y$) the latent dimension of the pretrained vision (resp. language) encoder. We consider the problem of learning alignment layers $f_{\theta_1} : \mathbb{R}^{d_x} \to \mathbb{R}^d$ and $g_{\theta_2} : \mathbb{R}^{d_y} \to \mathbb{R}^d$ that encode vision and language into a shared space of dimension $d$, parametrized by $\theta = (\theta_1, \theta_2)$.

In this setting, the training objective is often formulated as minimizing the divergence between the geometry of the shared space and a target geometry. Formally, given a dataset of image and language embeddings $X \in \mathbb{R}^{n \times d_x}$ and $Y \in \mathbb{R}^{n \times d_y}$, the goal is to minimize

$$\begin{aligned} \mathcal{L}(\theta; X, Y) = \\ \text{DIV}(K[f_{\theta_1}(X), g_{\theta_2}(Y)] \,\|\, K^*[X, Y]), \end{aligned} \tag{1}$$

where $K^*$ denotes the "target geometry" and DIV is some divergence between two affinity matrices. In the fully supervised setting, the dataset is made of pairs, i.e., $Y_i$ is the caption of $X_i$ and the target similarity is set to the identity

$$K^*[X, Y] = I_n. \tag{2}$$

For instance, the InfoNCE loss (Oord et al., 2018)

$$-\frac{1}{n} \sum_{i=1}^n \log \frac{\exp\Big(K[f(X), g(Y)]_{i,i}\Big)}{\sum_{j=1}^n \exp\Big(K[f(X), g(Y)]_{i,j}\Big)} \tag{3}$$

is a special case of (1) for $K^* = I_n$ and $\text{DIV}(K \,\|\, K^*) = \lim_{\varepsilon \to 0} \text{KL}(\text{Softmax}_\varepsilon(K^*) \,\|\, \text{Softmax}_1(K))$.

Thus, the main challenge to extend these approaches to unsupervised data is to introduce a target $K^*[X, Y]$ that is defined even if we *don't* assume that $X$ and $Y$ are pairs.

## 3.2. Semi-Supervised Setting

We consider a semi-supervised setting with three types of data. First, we observe a *small set of paired samples* $(A, B)$, where $A \in \mathbb{R}^{n_p \times d_x}$ and $B \in \mathbb{R}^{n_p \times d_y}$, and each row of $A$ is aligned with the corresponding row of $B$. In

---

**Algorithm 1** SOTAlign Training

**Require:** $(A, B), X, Y$
1: $(W_x, Wy) \leftarrow \text{LinearAlignment}(A, B)$
2: Initialize encoders $f$ and $g$
3: **for** $i = 1, \dots, T$ **do**
4:     Sample $X_b \sim X$   # Sample batch of image embeddings
5:     Sample $Y_b \sim Y$     # Sample batch of text embeddings
6:     $K^* \leftarrow \text{cosine}(X_b W_x^\top, Y_b W_y^\top)$
7:     $K \leftarrow \text{cosine}(f(X_b), g(Y_b))$
8:     $K_p \leftarrow \text{cosine}(f(A), g(B))$
9:     $\mathcal{L} \leftarrow \text{SigLIP}(K_p, I_{n_p}) + \alpha \text{KLOT}(K, K^*)$
10:    Update $f, g$ using $\nabla \mathcal{L}$
11: **end for**

---

addition, we have access to large collections of unpaired data: *unlabeled images* $X \in \mathbb{R}^{n_x \times d_x}$ and *unlabeled text* $Y \in \mathbb{R}^{n_y \times d_y}$. Finally, we assume that the number of supervised pairs is limited, i.e., $n_p \ll n_x$ and $n_p \ll n_y$. This setting is motivated by two considerations. First, it allows us to study how far supervision can be reduced while still enabling the recovery of a meaningful alignment between modalities, providing a direct test of the Platonic Representation Hypothesis. Second, such a regime reflects many practical scenarios in multimodal learning, where collecting paired data is expensive or infeasible and only a small number of aligned samples is available.

## 3.3. SOTAlign

We address this setting with a two step approach:

First, we fit a a simple *linear alignment model* with the supervised pairs $(A, B)$. We denote $W_x \in \mathbb{R}^{d' \times d_x}$ and $W_y \in \mathbb{R}^{d' \times d_y}$ the linear projections produced by this model. An important finding of this work is that such linear models already yield *surprisingly strong alignment*. We dedicate Section 4 to this fundamental component of our method.

Then we use this linear model to regularize the training of the final alignment layers $f_{\theta_1}$ and $g_{\theta_2}$ in a pseudo-labeling fashion. More precisely, we constrain the geometry of the learned shared space to stay close to that produced by the linear teacher. This regularization writes as

$$\begin{aligned} \Omega(\theta; X, Y) = \\ \text{DIV}\big(K\,[f_{\theta_1}(X), g_{\theta_2}(Y)]\big) \,\big\|\, K[XW_x^\top, YW_y^\top]\big), \end{aligned} \tag{4}$$

where the choice of the divergence DIV is the second core component of the method and is discussed in Section 5.

Finally, our training loss is

$$\mathcal{L}_\alpha(\theta; A, B, X, Y) = \mathcal{L}(\theta; A, B) + \alpha \Omega(\theta; X, Y), \tag{5}$$

where $\alpha$ controls the strength of the regularization.

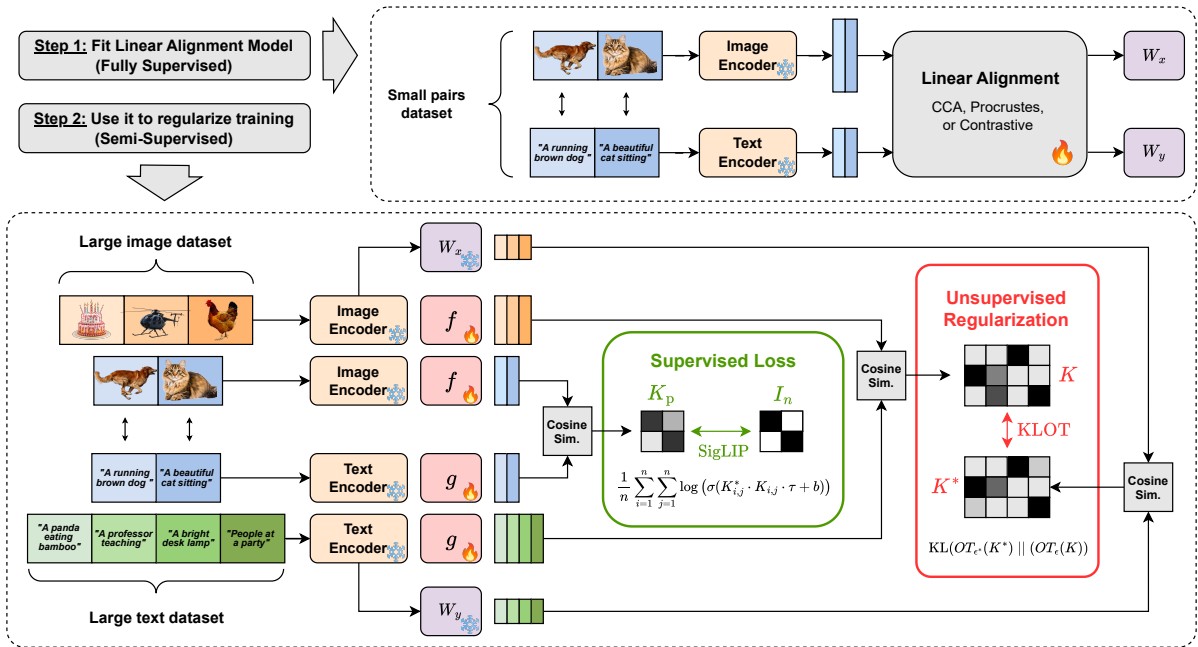

*Figure 2.* SOTAlign is a two-step method for the alignment of pretrained unimodal image and text encoders. First, we fit a linear alignment model only using the limited amount of available image-text pairs. Then, we use this linear model as a teacher to regularize the training of alignment layers $f$ and $g$ for a joint embedding space leveraging unimodal (unpaired) data.

## 4. Linear Alignment Model

The first core component of the proposed method is the linear alignment model. This model is trained with the limited amount of pairs available and then used as a teacher to regularize the training of the full semi-supervised model. As highlighted in the experimental section, such linear models achieve surprisingly strong alignment performances. We now discuss a selection of suitable candidates.

**Procrustes Alignment.** In the Orthogonal Procrustes problem, one tries to align two point clouds by looking for the orthogonal transformation that minimizes the RMSE (Schönemann, 1966). We slightly adapt its formulation to our setting by looking for two orthogonal transformations that map the pairs $(A, B)$ to a shared space. Formally,

$$(W_x, W_y) = \arg\max_{P,Q} \langle AP^\top, BQ^\top \rangle$$
$$\text{s.t.} \quad PP^\top = QQ^\top = I_{d'}. \quad (6)$$

This formulation assumes that the data is first centered and normalized which is omitted here for the sake of simplicity.

**Canonical Correlation Analysis.** In statistics, CCA is used to find a space in which two random variable are maximally correlated with each other (Mardia et al., 2024). Adapted to our setting, the two random variables are the text and image embeddings and CCA writes as

$$(W_x, W_y) = \arg\max_{P,Q} \langle AP^\top, BQ^\top \rangle$$
$$\text{s.t.} \quad (AP^\top)^\top (AP^\top) = (BQ^\top)^\top (BQ^\top) = I_{d'}. \quad (7)$$

The main difference from Procrustes is that the orthogonality constraint is applied to the shared space directions instead of the transformation itself. The solutions to Equation (6) and (7) are provided in Appendix C.1.

**Contrastive Learning.** Finally, perhaps the most natural choice is to consider a linear projection trained with a classical contrastive learning approach, formally

$$(W_x, W_y) = \arg\min_{P,Q} \text{DIV}\big(K[AP^\top, BQ^\top] \,\|\, I_{n_p}\big), \quad (8)$$

where DIV is the InfoNCE loss defined in equation (3) or an alternative such as the SigLIP loss (Zhai et al., 2023).

## 5. Choice of Divergence

The second core component of our method is the choice of a divergence $\text{DIV}(K \,\|\, K^*)$, where $K, K^* \in \mathbb{R}^{n \times n}$ denote, respectively, the affinity matrix induced by the trainable shared representation and the target affinity matrix. This section examines several possible choices for this divergence and discusses their respective strengths and limitations.

**Centered Kernel Alignment.** One of the most popular ways to compare affinity/kernel matrices is Centered Kernel Alignment (CKA) (Cristianini et al., 2001). Denoting $H = I_n - \mathbb{1}\mathbb{1}^\top$, CKA writes as

$$\text{CKA}(K, K^*) = \frac{\langle KH, HK^* \rangle}{\sqrt{\langle KH, HK \rangle \langle K^*H, HK^* \rangle}}. \quad (9)$$

CKA admits a linear-time computation in the batch size

$n$ when implemented via kernel factorizations (Proposition C.6), which constitutes a non-negligible practical property for alignment methods operating with large batches (Zhang et al., 2025a). However, CKA also suffers from known limitations (Davari et al., 2022) and, more importantly in our setting, enforces a strong constraint of the form $K \approx K^*$. This can be overly restrictive when $K^*$ is only intended as a regularizing signal provided by a linear teacher, rather than an exact target geometry.

**Generalized InfoNCE.** As highlighted above, it might be beneficial to use a regularization that does not enforce $K$ to be exactly aligned with $K^*$. To this end, one can consider the generalized InfoNCE loss (Shi et al., 2024) as it only enforces that $\arg\max_j K_{i,j} \approx \arg\max_j K_{i,j}^*$. This is achieved by first applying a Softmax on the affinity matrices before comparing them, i.e.,

$$
\begin{aligned}
&\text{InfoNCE}(K \parallel K^*) = \\
&\quad \text{KL}(\text{Softmax}_{\epsilon^*}(K^*) \parallel \text{Softmax}_\epsilon(K)).
\end{aligned}
\tag{10}
$$

The classical version is recovered for $\varepsilon^* \to 0$ and $K^* = I_n$.

**KLOT.** Given the previous interpretation of InfoNCE, we consider a natural extension that seeks the preservation of the entire Optimal Transport (OT) plan instead of only the nearest neighbor. Introducing $\Pi_n = \{ P \in \mathbb{R}_+^{n \times n} \mid P\mathbf{1} = \mathbf{1},\ P^\top\mathbf{1} = \mathbf{1} \}$, the OT plan is defined as

$$
OT_\epsilon(K) = \arg\min_{P \in \Pi_n} -\langle P, K \rangle + \epsilon H(P),
\tag{11}
$$

where $H(P) = \langle P, \log P \rangle$ is the negative entropy. Note that it is a natural extension of Softmax as $\text{Softmax}_\epsilon(K) = \arg\min_{P\mathbf{1}=\mathbf{1}} -\langle P, K \rangle + \epsilon H(P)$ and, similarly to Softmax, setting $\epsilon = 0$ recovers a strict one-to-one mapping. More details about OT are available in Appendix C.2.

Then, we define the KLOT divergence as

$$
\text{KLOT}(K \parallel K^*) = \text{KL}(OT_{\epsilon^*}(K^*) \parallel OT_\epsilon(K)).
\tag{12}
$$

This formulation is similar to that proposed by Van Assel et al. (2023) for the purpose of dimensionality reduction and generalizes (Shi et al., 2024) beyond $K^* = I_n$.

The main limitation of KLOT (5) is that $OT_\epsilon(K)$ does not admit a closed-form solution. While the Sinkhorn algorithm (Cuturi, 2013) provides fast convergence on GPUs, computing the gradient $\nabla_K OT_\epsilon(K)$ remains challenging. Existing approaches rely either on backpropagating through the Sinkhorn iterations by unrolling the algorithm (Genevay et al., 2018), which induces a severe memory bottleneck, or on implicit differentiation techniques (Eisenberger et al., 2022), which significantly increase time complexity. We *fully address* this limitation by deriving an explicit expression for the gradient (Theorem 5.1).

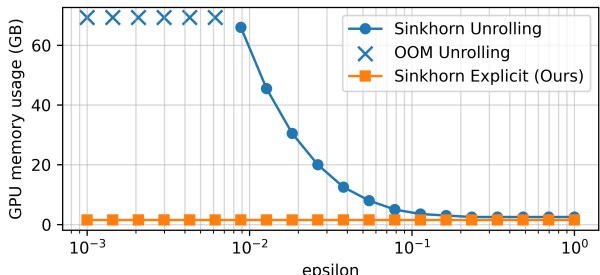

*Figure 3.* GPU memory usage for a batchsize $n = 10k$ when computing the gradient of the OT-based divergence with naive solver unrolling (blue) and the provided explicit gradient formula (orange). Additional results are reported in Appendix B.1.

**Theorem 5.1.** *The gradient of the KLOT divergence is,*

$$
\nabla_K \text{KLOT}(K \parallel K^*) = \frac{OT_\epsilon(K) - OT_{\epsilon^*}(K^*)}{\epsilon}.
\tag{13}
$$

*Proof is provided in Appendix C.2. We also refer to this blog post (Assel, 2024), which draws an insightful connection to the Monge Gap regularizer (Uscidda & Cuturi, 2023).*

As illustrated in Figure 3, our approach removes the memory bottleneck inherent to Sinkhorn unrolling and can be up to $50\times$ faster than implicit differentiation (Figure 8). This is a general result that could potentially apply to a range of OT-based methods using similar objectives, including recent approaches in model alignment and contrastive learning (Van Assel et al., 2023; Mo et al., 2023; Shi et al., 2024).

# 6. Experiments

**Experimental Setting.** We train all models using a maximum batch size of 32k, composed of up to 10k paired samples and completed with unpaired images and text. We use the LION optimizer (Chen et al., 2023) with a cosine annealing learning-rate schedule, a maximum learning rate of $10^{-4}$, and a weight decay of $10^{-5}$, and train for 2000 iterations. For the supervised component of the loss, we employ the SigLIP objective, initializing the logit scale to 20 and the logit bias to $-10$, with both parameters learned during training. Unless otherwise specified, we use DINOv3 ViT-L (Siméoni et al., 2025) and NV-Embed-v2 (Lee et al., 2025) as the pretrained vision and language encoders, respectively. By default, experiments are conducted with 10k paired samples and, when applicable, up to 1M *unpaired* images and texts drawn from CC3M (Sharma et al., 2018). We vary the weight of the regularization in Equation (5) over $\alpha \in \{10^{-3}, 10^{-4}, 10^{-5}\}$ and select it based on retrieval performance on CC3M, accounting for different ratios of supervised to unsupervised data. We show in Section 6.2 that comparable results can be obtained with alternative settings. By default, the metric

*Table 1.* Comparison of linear methods across different divergences. Zero-shot retrieval on COCO (MeanR@1). "None" indicates the standalone performances of the linear method.

| Divergence | Linear Method | | |
| --- | --- | --- | --- |
| | Procrustes | CCA | Contrastive |
| None | 21.1 | 21.5 | 24.2 |
| CKA | 23.5 | 23.5 | 24.2 |
| InfoNCE | 23.9 | 24.1 | 26.5 |
| KLOT | **30.0** | **30.3** | **28.5** |

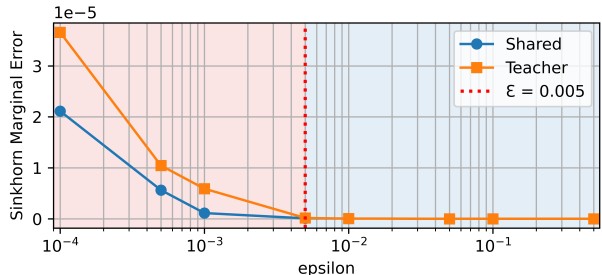

*Figure 4.* We plot Sinkhorn convergence criterion $||P\mathbf{1}-\mathbf{1}||_1$ after 100 iterations against $\epsilon$. The cost matrix of size 32k used for this plot is extracted from a SOTAlign training batch.

that we report is the average of the text-to-image (T2I) and image-to-text (I2T) retrieval (Recall@1) performance on the COCO validation set which we denote MeanR@1.

## 6.1. Design Choices

**Linear Methods.** SOTAlign employs a linear teacher which can be fit using the methods described in Section 4. We report the standalone zero-shot retrieval performance of these linear models when trained on only 10k image-text pairs from CC3M in the first row of Table 1. Notably, even the closed form models (CCA and Procrustes) reach MeanR@1 scores larger than 21%. The linear contrastive approach (SigLIP loss, SAIL baseline) reaches 24.2% and will serve as the main supervised baseline.

**Divergences.** To align the teacher's affinity matrix with the affinity matrix in the learnable shared space, SOTAlign can utilize any of the three divergences detailed in Section 5. Table 1 displays all combinations of linear methods and divergences. Critically, we observe that the proposed OT-based divergence KLOT systematically outperforms the classical alternatives. The best results is achieved for CCA and KLOT and we select this setting for the next experiments.

**Entropic Regularization.** Choosing the entropic regularization parameters $\epsilon$ and $\epsilon^*$ is a classical challenge when using the Sinkhorn algorithm. Fortunately, we identify two simple guidelines that avoid expensive grid search. First, we want the teacher to produce a sharp signal, so $\epsilon^*$ should be set to the smallest value that ensures Sinkhorn convergence within a fixed iteration budget. For the budget of 100 iterations used in this work, this yields $\epsilon^* = 0.005$ (Figure 4). Second, we set $\epsilon > \epsilon^*$ so that the student distribution has higher entropy than the teacher, encouraging it to sharpen toward the teacher's geometry. In the experiments, we set $\epsilon/\epsilon^* = 10$ as we find that this simple choice yields consistently optimal results across supervision levels (Figure 12).

## 6.2. Impact of Supervision Regime

**Number of Supervised Pairs.** We first analyze how the number of paired image–text samples affects downstream performance. In this experiment, we fix the amount of unpaired data to 1M images and 1M texts from CC3M, and vary the number of paired samples from $10^2$ to $10^5$. Figure 5 reports zero-shot retrieval results. Across all supervision levels, SOTAlign consistently outperforms the supervised SAIL baseline, with gains of up to $+10\%$ accuracy in the intermediate regime between $10^3$ and $10^4$ pairs. As expected, these gains diminish as the number of paired samples increases, and both methods fail under extremely sparse supervision (100 pairs). Overall, SOTAlign reaches the same performances as SAIL with roughly 4 times less supervision.

**Number of Unsupervised Samples.** Next, we investigate the effect of unpaired data on downstream performance. In this experiment, we fix the number of pairs to 10k, and vary the number of additional unpaired image and text samples between $10^4$ and $10^6$. Figure 5 shows that our method successfully leverages unpaired data for zero-shot retrieval with consistent gains up to 500k unpaired samples.

## 6.3. Impact of Data Source and Domain Shift

**Unsupervised Data Source.** We further evaluate our method in a challenging cross-dataset regime where unpaired images and texts originate from entirely different sources (Table 9). Using a fixed set of 10k paired samples from CC3M, we introduce unpaired unimodal data from CC12M, COCO, and ImageNet-1k, as well as synthetic captions. Despite these shifts in the data distributions, our approach consistently outperforms the supervised baseline. Notably, incorporating ImageNet-1k images improves classification performance, while leveraging COCO samples yields retrieval gains by narrowing the gap to the test distribution. These results demonstrate that our framework can effectively exploit unpaired data even when the visual and textual modalities are drawn from disjoint, heterogeneous corpora.

**Quantifying the Distribution Shift.** Motivated by these results, we next seek to quantify the effect of distribution shift and relate it to the observed performance gains. Given

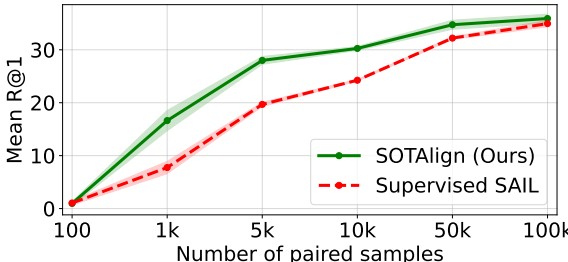 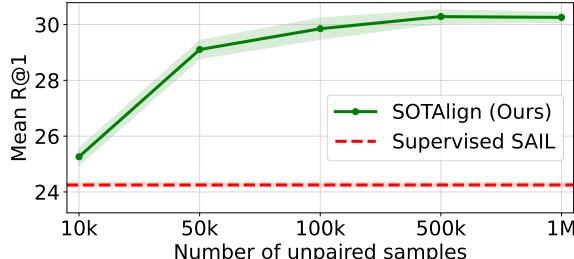

*Figure 5.* Left: Effect of the number of paired samples (while fixing 1M unpaired samples). Right: Effect of the number of unpaired samples (while fixing 10k pairs). We report the zero-shot retrieval (MeanR@1) on COCO. More metrics are reported in Appendix B.

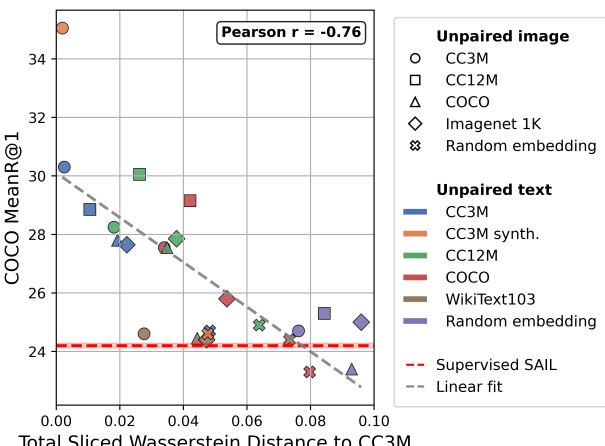

*Figure 6.* Relationship between the total sliced Wasserstein distance between CC3M image/text dataset and unimodal datasets, and the downstream performance of SOTAlign trained on 10k CC3M image–text pairs and up to 1M samples from the corresponding unimodal datasets. As a baseline, we consider random embeddings sampled uniformly on the hyper-sphere.

a source of unpaired data $\mathcal{D} = (X, Y)$ and the paired dataset $\mathcal{D}_p = (A, B)$, we define the distribution shift as

$$\mathrm{d}(\mathcal{D}, \mathcal{D}_p) = \mathrm{SSW}(X, A) + \mathrm{SSW}(Y, B), \quad (14)$$

where SSW denotes the Spherical Sliced Wasserstein distance (Liu et al., 2024), with details in Appendix C.2. We adopt this metric because it is scalable to large datasets, well suited to unit-normalized embeddings, and does not require $X$ and $A$ (or $Y$ and $B$) to be aligned. As shown in Figure 6, this distance strongly correlates (Pearson $r = -0.76$) with downstream performance: unpaired data that are closer to the paired distribution consistently yield larger performance gains when incorporated during training. To probe extreme distribution shift, we replace the unpaired images and text with Gaussian-sampled embeddings. In this setting, the retrieval performance remains close to the supervised SAIL baseline, indicating that even severely mismatched unpaired data do not substantially degrade alignment.

*Table 2.* SOTAlign compared to supervised SAIL across different paired datasets (10k pairs) and 1M unpaired samples from CC3M. Zero-shot retrieval on COCO (MeanR@1). Green values (+) represent the absolute gain.

| Paired data | Method | ImageNet 1K | COCO T2I | COCO I2T |
|---|---|---|---|---|
| CC3M | SAIL | 35.6 | 21.0 | 27.4 |
| | SOTAlign | 46.1 +10.5 | 26.5 +5.5 | 34.1 +6.7 |
| CC3M synthetic | SAIL | 36.2 | 28.3 | 37.2 |
| | SOTAlign | 46.5 +10.3 | 31.3 +3.0 | 43.1 +5.9 |
| CC12M | SAIL | 38.5 | 20.1 | 27.2 |
| | SOTAlign | **47.4** +8.9 | 26.1 +6.0 | 36.3 +9.1 |
| COCO | SAIL | 21.8 | 30.7 | 42.4 |
| | SOTAlign | 35.8 +14.0 | **34.8** +4.1 | **46.7** +4.3 |

**Supervised Data Source.** We next study the impact of the paired data source on SOTAlign performance (Table 2), while fixing the unpaired data to 1M samples from CC3M. Varying the source of the 10k image–text pairs reveals that higher-quality supervision can substantially influence alignment. In particular, using CC3M pairs with synthetic captions yields a notable improvement in retrieval performance (+4.8% T2I R@1), suggesting that cleaner textual supervision better guides the exploitation of noisy unpaired data. While pairs drawn from the larger CC12M corpus improve ImageNet classification, the strongest retrieval performance is obtained when using COCO pairs.

### 6.4. Impact of Unimodal Encoders

In the same vein, we examine the impact of the choice of unimodal encoders on zero-shot classification and retrieval performance. We fix the training data to the standard setting of 10k paired samples and 1M unpaired samples from CC3M, and vary only the vision and language encoders. As reported in Table 3, aligning DINOv3 ViT-L with NV-Embed-v2 yields the strongest downstream performance, achieving 46.1% accuracy on ImageNet and 26.5% T2I R@1 on COCO. This trend aligns with the *Platonic Representation Hypothesis* (Huh et al., 2024), which suggests that

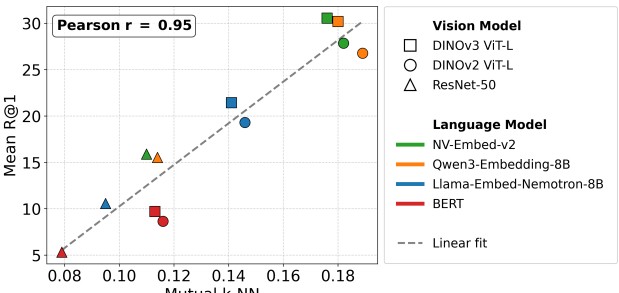

*Figure 7.* Relationship between the representational similarity (mutual kNN) of frozen encoder pairs and their retrieval performance after semi-supervised alignment using SOTAlign.

*Table 3.* SOTAlign trained on 10k pairs and 1M unpaired samples from CC3M with different unimodal encoders. Zero-shot classification (top-1 accuracy) and retrieval (Recall@1). Green values represent the absolute gain over supervised SAIL.

| Vision Model | Language Model | ImageNet 1K | COCO T2I | COCO I2T |
|---|---|---|---|---|
| DINOv2 | Nemotron-8B | 32.4 +6.9 | 15.5 +3.8 | 23.3 +5.3 |
| | Qwen3-8B | 39.5 +7.7 | 20.9 +4.1 | 31.1 +7.3 |
| | NV-Embed-v2 | 42.5 +9.8 | 23.1 +4.1 | 31.1 +7.7 |
| DINOv3 | Nemotron-8B | 35.5 +10.1 | 16.6 +3.8 | 26.2 +5.2 |
| | Qwen3-8B | 42.7 +9.3 | 24.1 +5.0 | 35.3 +7.3 |
| | NV-Embed-v2 | 46.1 +10.5 | 26.5 +5.5 | 34.1 +6.7 |

*Table 4.* Zero-shot image-text retrieval (Recall@1) on COCO and Flickr30. Comparison of supervised and semi-supervised methods trained with 10k image-text pairs and 1M unpaired samples from CC3M. Upper bound with 1M supervised pairs in grey.

| | Method | COCO T2I | COCO I2T | Flickr30k T2I | Flickr30k I2T |
|---|---|---|---|---|---|
| Sup. | SAIL (1M) | 35.5 | 45.5 | 63.1 | 75.0 |
| | SAIL | 21.0 | 27.4 | 45.7 | 54.1 |
| | STRUCTURE | 21.0 | 28.7 | 46.8 | 54.0 |
| Semi-sup. | SAIL | 20.7 | 26.5 | 44.9 | 53.1 |
| | STRUCTURE | 20.9 | 28.0 | 45.7 | 56.0 |
| | NNCLR | 21.3 | 27.9 | 46.6 | 53.0 |
| | S-CLIP | 20.4 | 27.8 | 44.5 | 52.6 |
| | **SOTAlign (Ours)** | **26.5** | **34.1** | **51.7** | **60.8** |

*Table 5.* Zero-shot image classification (top-1 accuracy). Comparison of supervised and semi-supervised methods trained with 10k image-text pairs and 1M unpaired samples from CC3M. Upper bound with 1M supervised pairs in grey.

| | Method | Food-101 | CIFAR-10 | CIFAR-100 | DTD | ImageNet |
|---|---|---|---|---|---|---|
| Sup. | SAIL (1M) | 63.9 | 97.8 | 82.3 | 53.5 | 56.4 |
| | SAIL | 36.4 | 96.2 | 71.2 | 36.8 | 35.6 |
| | STRUCTURE | 38.5 | 96.7 | 72.2 | 39.5 | 38.2 |
| Semi-sup. | SAIL | 36.5 | 96.2 | 71.3 | 35.9 | 35.6 |
| | STRUCTURE | 37.6 | 96.5 | 70.8 | 38.7 | 36.8 |
| | NNCLR | 37.9 | 96.5 | 73.0 | 38.8 | 37.4 |
| | S-CLIP | 35.3 | 95.9 | 69.3 | 37.6 | 36.4 |
| | **SOTAlign (Ours)** | **50.0** | **97.5** | **78.3** | **42.4** | **46.1** |

as models scale in data and capacity, their representation spaces increasingly converge. DINOv3's substantially larger pretraining corpus (1.7 billion images, versus 142 million for DINOv2) likely explains its consistent advantage.

We then extend our analysis to weaker encoders, including ResNet-50 (He et al., 2016) for images and BERT (Devlin et al., 2019) for captions, and measure the representational similarity between frozen vision and text embeddings using mutual $k$-NN, a parameter-free metric that quantifies the overlap in nearest-neighbor structure across modalities prior to any training. As shown in Figure 7, we observe a striking correlation (Pearson $r = 0.95$) between mutual $k$-NN and downstream Mean R@1, across all encoder pairs considered. This result has a practical implication: mutual $k$-NN serves as a reliable and fast diagnostic to predict whether a given pair of encoders is compatible before running SOTAlign.

### 6.5. Benchmarking Semi-Supervised Alignment

**Baselines.** Since the semi-supervised alignment setting we consider is relatively unexplored, there are no established standard baselines. We therefore compare SOTAlign against a range of supervised and semi-supervised methods which we adapt to our setting. The primary supervised baseline is SAIL (Zhang et al., 2025a), trained on paired image–text

data with a SigLIP loss; we also propose a semi-supervised variant that incorporates unpaired samples as additional negatives. We further consider STRUCTURE (Gröger et al., 2025), which regularizes joint embeddings to preserve unimodal geometry, and evaluate this term using either paired data only or both paired and unpaired samples. In addition, we include pseudo-labeling approaches that construct synthetic pairs from similarity distributions, including NNCLR (Dwibedi et al., 2021) (as used in DeCLIP (Li et al., 2022)) and S-CLIP (Mo et al., 2023). Finally, we compare against SUE (Yacobi et al., 2025), a semi-supervised alignment method restricted to retrieval on a single dataset. Full details of all baselines are provided in Appendix A.2.

**Zero-Shot Image-Text Retrieval.** We evaluate SOTAlign against these baselines in T2I and I2T retrieval on COCO (Lin et al., 2014) and Flickr30k (Plummer et al., 2015), and report the results in Table 4. In the low-resource regime with 10k image-text pairs, the supervised baseline SAIL reaches 21.0 T2I R@1 and 27.4 I2T R@1. STRUCTURE performs marginally better benefiting from its structure-preservation objective. However, both methods fail to exploit unpaired

*Table 6.* Alignment per dataset. Following the setup of SUE (Yacobi et al., 2025), we train for alignment per dataset and evaluate image-text retrieval (Recall@5).

| Method | COCO 100 Pairs | | Flickr30k 500 Pairs | | Polyvore 500 Pairs | |
|---|---|---|---|---|---|---|
| | I2T | T2I | I2T | T2I | I2T | T2I |
| CSA | 1.3 | 1.0 | 1.3 | 0.8 | 1.3 | 1.0 |
| Contrastive | 8.5 | 5.8 | 9.5 | 9.8 | 13.8 | 11.5 |
| SUE | 21.5 | 18.3 | 19.8 | 22.0 | 22.8 | 20.8 |
| **SOTAlign (Ours)** | **35.8** | **35.0** | **59.8** | **63.3** | **55.3** | **55.3** |

*Table 7.* Generalization across domains and modalities. We report R@10 on SkyScript satellite image-text retrieval (10k test pairs) and R@1 on LibriSpeech speech-text retrieval (2.6k test pairs).

| Method | SkyScript | | LibriSpeech | |
|---|---|---|---|---|
| | T2I | I2T | S2T | T2S |
| CCA | 21.0 | 21.3 | 72.4 | 75.1 |
| SAIL | 24.4 | 25.7 | 55.8 | 65.5 |
| STRUCTURE | 26.3 | 26.9 | 65.4 | 68.1 |
| **SOTAlign (Ours)** | **27.3** | **27.4** | **78.4** | **83.1** |

data, either as additional negatives or for structure preservation. Notably, even the adapted semi-supervised approaches, NNCLR and S-CLIP, are unable to successfully exploit unpaired data. S-CLIP has been originally developed for domain adaptation and appears less robust when confronted with the large diversity of unpaired samples in our setting. Its pseudo-labels are further limited to the small set of paired instances. In contrast, SOTAlign successfully leverages the 1M unpaired images and texts from CC3M to improve cross-modal alignment. On Flickr30k, our method reaches 51.7 T2I R@1 and 60.8 I2T R@1, yielding gains of +4.9 and +4.8 over the strongest baselines, respectively. For comparison, we include in gray the supervised upper bound obtained by training SAIL with 1M paired examples.

**Zero-Shot Image Classification.** We further evaluate SOTAlign in zero-shot classification on ImageNet (Deng et al., 2009) and more fine-grained classification datasets. The results are displayed in Table 5 and mirror the trends observed in zero-shot retrieval. STRUCTURE outperforms SAIL in the supervised setting, but is not able to leverage unpaired data for additional performance gains. Existing semi-supervised methods like NNCLR and S-CLIP do not show any improvements over the supervised baselines. Only SOTAlign is able to leverage 1M unpaired samples during alignment to improve zero-shot image classification. Our method achieves an ImageNet top-1 accuracy of 46.1%, which is an improvement of +7.9 over the best baseline. In grey we display the supervised SAIL baseline trained on 1M image-text pairs as an upper bound.

**Alignment per Dataset.** Yacobi et al. (2025) study semi-supervised vision–language alignment using pretrained encoders, but under a substantially simpler setting than ours: their method operates within a single dataset, with paired and unpaired samples drawn from the same distribution and evaluation restricted to retrieval on small test splits (400 samples). In contrast, our setting involves cross-dataset unpaired data and multiple downstream tasks. Nevertheless, when evaluated in their setting, SOTAlign consistently outperforms SUE (Yacobi et al., 2025) and its baselines, achieving gains of +14.3 I2T R@5 on COCO, +40.0 on Flickr30k, and +32.5 on Polyvore (see Table 6).

**Generalization across Domains and Modalities.** While our main experiments focus on general-purpose image-text alignment, we further investigate whether SOTAlign generalizes to domain-specific data and to modalities beyond vision and language. First, we evaluate SOTAlign in the field of remote sensing using satellite images and multi-object captions from SkyScript (Wang et al., 2024). Specifically, we train SOTAlign on 10k pairs augmented with 700k unpaired image and text samples, and assess zero-shot retrieval on 10k test pairs. As shown in Table 7, SOTAlign reaches Mean R@10 = 27.4 in this target domain, surpassing STRUCTURE and significantly outperforming SAIL by +2.3%. Additional results on SkyScript across varying supervision and evaluation regimes are provided in Appendix B.4. Second, we examine speech-text alignment on LibriSpeech (Panayotov et al., 2015), using WavLM-Large (Chen et al., 2022) as speech encoder and NV-Embed-v2 as text encoder. We train SOTAlign on 1k pairs augmented with 100k unpaired speech and text samples, and evaluate zero-shot retrieval on the test set with 2.6k pairs. Table 7 shows that SOTAlign substantially outperforms all baselines, improving Mean R@1 by +14.0 over STRUCTURE and by +20.1 over SAIL. These results suggest that SOTAlign provides an effective framework for semi-supervised alignment across domains and modalities.

## 7. Conclusion

In this work, we introduced a semi-supervised setting for aligning pretrained unimodal encoders, which we believe is relevant to many real-world modalities where large-scale paired data are scarce. We argue that vision–language alignment provides an ideal testbed for this problem, as abundant paired data enable systematic exploration of different supervision regimes. To the best of our knowledge, SOTAlign is the first model that can effectively leverage large-scale unimodal data for multimodal alignment in this setting. We hope that the simplicity of SOTAlign will inspire future work on multimodal representation alignment beyond fully supervised regimes.

## Acknowledgments

This work was partially funded by the ERC (853489 - DEXIM) and the Alfried Krupp von Bohlen und Halbach Foundation, which we thank for their generous support. This work was also supported by Hi! PARIS and ANR/France 2030 program (ANR-23-IACL-0005) and by the French National Research Agency (ANR) through the France 2030 program under the MacLeOD project (ANR-25-PEIA-0005). Finally, it received funding from the Fondation de l'École polytechnique. We are grateful to Rémi Flamary, Florence d'Alché-Buc, and Charlotte Laclau for their helpful comments.

## Impact Statement

This work aims to advance research in machine learning, particularly the study of multimodal representation alignment. While improved alignment methods may have broad downstream applications, we do not identify any specific societal impacts that require explicit discussion here.

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

# A. Experimental Setting

We outline the experimental setup in Section 6.5. Here, we provide further details on our implementation and baselines.

## A.1. Implementation Details

Following Zhang et al. (2025a), we create global image representations by concatenating the `[CLS]` token with the mean of the remaining patch tokens. Text features are computed by averaging all patch tokens. We project both modalities into a shared embedding space of dimensionality $d = 1024$ using linear layers $f$ and $g$. We found them to be more robust in our low-supervision regime compared to non-linear layers.

When performing CCA, we add a regularization $\lambda = 0.1$ to the eigenvalues of matrices that need to be inverted. Our divergence, KLOT, is computed using the Sinkhorn algorithm with $n = 100$ iterations in both spaces. We set the entropic regularization to $\epsilon = 0.01$ in the reference space and $\epsilon = 0.05$ in the joint embedding space.

Our experiments are conducted with 10k paired samples and, when applicable, up to 1M unpaired images and texts drawn from CC3M (Sharma et al., 2018). We train all models using a maximum batch size of 32k, composed of up to 10k paired samples and completed with unpaired images and text. If there are less than 32k total samples available in our robustness studies, we adjust the batch size accordingly. We use the LION optimizer (Chen et al., 2023) with a cosine annealing learning-rate schedule, a maximum learning rate of $10^{-4}$, and a weight decay of $10^{-5}$, and train for 2000 iterations.

We mainly employ DINOv3 ViT-L (Siméoni et al., 2025) and NV-Embed-v2 (Lee et al., 2025) as the pretrained vision and language encoders, respectively. In Section 6.4, we additionally evaluate SOTAlign with DINOv2 ViT-L (Oquab et al., 2024), Qwen3-Embedding-8B (Zhang et al., 2025b), and Llama-Embed-Nemotron-8B (Babakhin et al., 2025). All of these language models are among the top performing models in the MMTEB benchmark (Enevoldsen et al., 2025).

Our main evaluation metric is the average of the text-to-image (T2I) and image-to-text (I2T) retrieval (Recall@1) performance on the COCO validation set which we denote MeanR@1. Whenever required, we use a similar score on the CC3M validation split for hyperparameter selection.

All experiments can be run on a single A100 GPU with 80 GB memory.

## A.2. Baselines

In Section Section 6.5, we compare SOTAlign against several supervised and semi-supervised baselines in zero-shot image classification and retrieval. For each baseline, we consider various configurations as detailed below, and report their optimal performance after hyperparameter tuning.

**SAIL** (Zhang et al., 2025a) performs contrastive learning of alignment layers with the SigLIP (Zhai et al., 2023) loss exclusively on paired data. This method represents a series of recent supervised contrastive methods for the alignment of pretrained unimodal vision and language models (Vouitsis et al., 2024; Maniparambil et al., 2025; Huang et al., 2025). Following Zhang et al. (2025a), we initialize the logit scale to 20 and the logit bias to $-10$ and allow both parameters to be trained. We examine the extension of SAIL to our semi-supervised setting by incorporating unpaired samples as additional negatives in the SigLIP loss.

**STRUCTURE** (Gröger et al., 2025) aligns pretrained encoders in low-resource regimes by augmenting the contrastive objective with an additional loss that forces the similarity distribution in the joint embedding space to lie between the unimodal similarity distributions. While STRUCTURE focuses on a fully supervised setting with paired data, we also evaluate the strength of its regularization term on unpaired data in our semi-supervised setting. We set the number of levels to 1 and the temperature in the softmax function to $\tau = 0.07$. We tune the weight of the structure preservation term over $\lambda \in \{0.1, 1, 10, 100, 1000\}$, and consider both no warmup and a 500-step warmup schedule.

Further semi-supervised techniques can be borrowed from contrastive pretraining and low-resource domain adaptation to construct pseudo-pairs based on similarity distributions in the unimodal or joint embedding spaces.

**NNCLR** (Dwibedi et al., 2021) enriches contrastive learning by retrieving the nearest-neighbors of an instance and using them as additional positives. DeCLIP (Li et al., 2022) has adopted such nearest-neighbor supervision in image-language pre-training. We follow this line of work and utilize unpaired images and text as augmentation for the few paired samples. Specifically, for a given image, we find the closest neighbor of its paired caption in the unimodal language space, which then

serves as an additional positive for the image. Similarly, for a given text, we find the closest neighbor of its paired image in the unimodal vision space, and can use it as an additional positive for the text. While NNCLR is often implemented with a queue containing the last few batches during training, we can compute the nearest neighbors for all CC3M samples in the unimodal spaces a priori since we utilize pretrained encoders. When training the alignment layers, we then randomly sample a nearest neighbor from the top $k \in \{1, 5, 10\}$ neighbors, and further perform hyperparameter search for the weights of the contrastive losses with the additional positives: $w_{\mathrm{img}}, w_{\mathrm{text}} \in \{0, 0.1, 1, 10\}$.

**S-CLIP** (Mo et al., 2023) addresses the domain adaption of CLIP with pseudo-labeling at the caption and keyword level. We evaluate their caption-level supervision in our setting. Given an unpaired image, S-CLIP computes similarity scores to paired images, and then uses the resulting similarity distribution to determine pseudo positives from the paired text. The pseudo-positives can be chosen in a hard assignment as the single nearest neighbor (argmax of the distribution, similar to NNCLR) or in a soft assignment as a weighted average of representations. A key component of S-CLIP is its use of OT to find the optimal matching between unpaired and paired images. The method is limited by the small pool of positives. We apply S-CLIP in the unimodal vision and language spaces as well as in the joint-embedding space. We search for pseudo-labels for both unpaired images as well as unpaired text and tune their corresponding weights in the final objective via a grid search: $w_{\mathrm{img}}, w_{\mathrm{text}} \in \{0, 0.1, 1, 10\}$.

**SUE** (Yacobi et al., 2025) studies the alignment of unimodal encoders on a single image-text dataset. Their approach combines learnable spectral embeddings on unpaired data, with CCA on paired data for linear alignment, and a residual network to further refine the alignment.

### A.3. Datasets

We construct our semi-supervised setting primarily using CC3M (Sharma et al., 2018) with both raw web captions and synthetic captions generated by DreamLIP (Zheng et al., 2024). We further experiment with disjoint images and texts from CC12M (Changpinyo et al., 2021), COCO (Lin et al., 2014), ImageNet (Deng et al., 2009), and WikiText (Merity et al., 2016). We select models based on their average text-to-image (T2I) and image-to-text (I2T) retrieval performance on the CC3M validation set.

In Section 6.5, we evaluate our model in a zero-shot setting across a diverse suite of classification and retrieval benchmarks.

For **classification** we consider the following datasets: (Deng et al., 2009), Food-101 (Bossard et al., 2014), CIFAR-10 (Krizhevsky, 2009), CIFAR-100 (Krizhevsky, 2009), Aircraft (Maji et al., 2013),DTD (Cimpoi et al., 2014) and Flowers (Nilsback & Zisserman, 2008).

For **retrieval** we consider: COCO (Lin et al., 2014), Flickr30 (Plummer et al., 2015), SkyScript (Wang et al., 2024) and LibriSpeech (Panayotov et al., 2015).

*Figure 8.* Comparison of memory usage, runtime, and number of Sinkhorn iterations for different gradient computation strategies. We run as many Sinkhorn iterations as required to achieve marginal convergence within a tolerance of $10^{-6}$.

# B. Additional Experiments

## B.1. Sinkhorn Backpropagation

The Sinkhorn algorithm has recently been used as a differentiable layer in a wide range of applications, including reinforcement learning (Emami & Ranka, 2018), learning to rank (Adams & Zemel, 2011), discriminant analysis (Flamary et al., 2018), graph matching (Krzakala et al., 2025), and representation learning (Van Assel et al., 2023). Closer to our setting, several recent works have applied Sinkhorn-based objectives to contrastive learning for vision–language models (Mo et al., 2023; Shi et al., 2024).

While the Sinkhorn algorithm (defined in Appendix C.2) is differentiable in theory, computing its gradient in practice is challenging. The most common approach consists in *unrolling* the Sinkhorn iterations and directly backpropagating through the solver. However, this strategy incurs a large memory overhead, as the full computational graph must be retained for all iterations, causing memory consumption to grow rapidly with the number of Sinkhorn steps.

An alternative is to rely on implicit differentiation, which amounts to solving the linear system defined by the optimality conditions of the entropic OT problem. In the case of Sinkhorn, this system exhibits a particular structure that enables more efficient solvers (Cuturi et al., 2020; Eisenberger et al., 2022). While this approach alleviates the memory explosion associated with unrolling, it remains computationally expensive in practice.

In the context of our proposed divergence,

$$\text{KLOT}(K \,||\, K^*) = \text{KL}(OT_{\epsilon^*}(K^*) \,||\, OT_\epsilon(K)), \tag{15}$$

a naive application of the chain rule would suggest that computing the gradient

$$\nabla_K \text{KLOT}(K \,||\, K^*)$$

requires explicitly forming the Jacobian

$$\frac{\partial \, OT_\epsilon(K)}{\partial K},$$

thereby necessitating either Sinkhorn unrolling or implicit differentiation.

Crucially, Theorem 5.1 shows that this is not required. Instead, the gradient of KLOT admits a closed-form expression that can be computed directly, without evaluating the Jacobian of the Sinkhorn operator.

To empirically illustrate the efficiency of this result, we extract an $n \times n$ affinity matrix with $n = 10k$ from a checkpoint of SAIL training and compare three strategies for computing $\nabla_K \text{KLOT}(K \,||\, K^*)$: Sinkhorn unrolling, implicit differentiation, and our closed-form gradient. The results are reported in Figure 8. Our approach significantly outperforms both alternatives in terms of memory usage and runtime. In particular, depending on the value of $\epsilon$, which controls the number of Sinkhorn iterations (with convergence scaling as $\mathcal{O}(1/\epsilon^2)$), the proposed method can be up to $100\times$ more memory efficient than unrolling and up to $50\times$ faster than implicit differentiation.

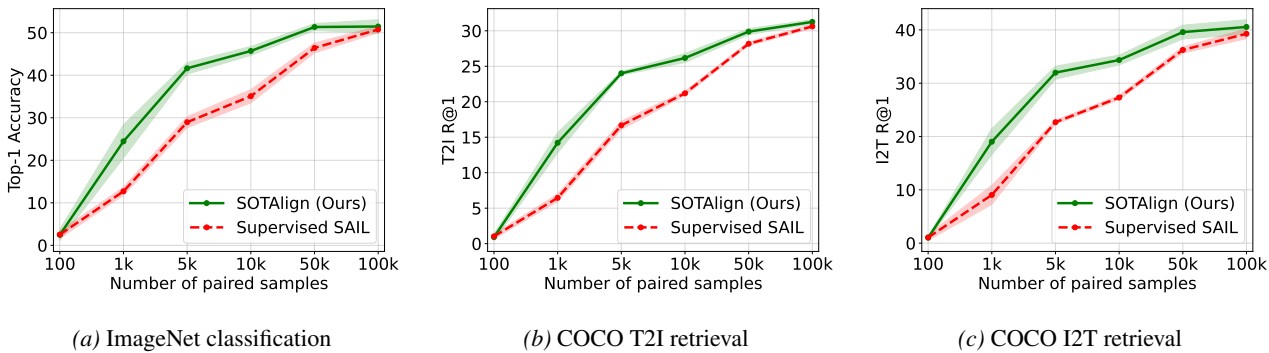

*(a)* ImageNet classification       *(b)* COCO T2I retrieval       *(c)* COCO I2T retrieval

*Figure 9.* Effect of the number of paired samples during alignment on downstream zero-shot classification and retrieval. We fix 1M unpaired samples from CC3M and vary the number of paired samples.

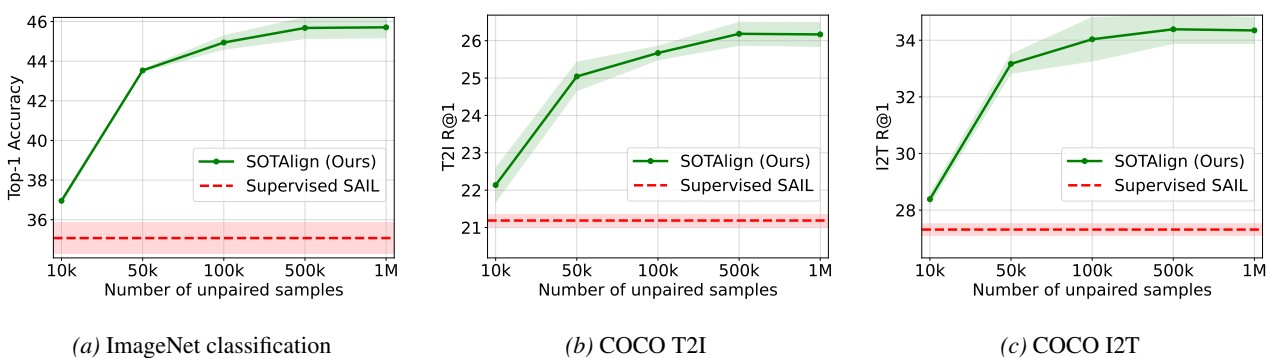

*(a)* ImageNet classification       *(b)* COCO T2I       *(c)* COCO I2T

*Figure 10.* Effect of the number of unpaired samples during alignment on downstream zero-shot classification and retrieval. We fix 10k paired samples from CC3M and vary the number of unpaired samples.

## B.2. Robustness of SOTAlign

**Number of Samples.** In Section 6.2 and Section 6.3, we analyze the robustness of SOTAlign to variations in both the amount and the source of supervised and unsupervised data. Here, we report the full set of results. Figure 9 shows how zero-shot classification and retrieval performance vary as a function of the number of paired samples used during alignment. Figure 10 illustrates the effect of increasing the number of unpaired samples, in comparison to the supervised SAIL baseline. Table 10 reports zero-shot classification and retrieval results for different combinations of unimodal vision and language encoders, along with absolute gains over supervised SAIL.

**Teacher Quality.** We next evaluate the sensitivity of SOTAlign to the quality of the linear teacher. To this end, we randomly shuffle a fraction $p$ of the image-text pairs used for teacher training, thereby injecting $p \times 100\%$ of incorrect pairs in the training data. As shown in Table 8, SOTAlign remains substantially above the supervised SAIL baseline even when half of the pairs are corrupted ($p = 0.5$). We attribute this robustness to the limited capacity of the linear teacher (which prevents overfitting to noisy pairs) and to the natural invariance of the optimal transport loss.

**Weight of OT Loss.** We further analyze the sensitivity of SOTAlign to the regularization weight $\alpha$, which controls the contribution of the KLOT divergence on unpaired data to the overall loss. First, we study this sensitivity across different supervision regimes. Figure 11a reports a grid search over $\alpha \in \{10^{-6}, 10^{-5}, 10^{-4}, 10^{-3}, 10^{-2}\}$ for varying numbers of paired samples. The default value $\alpha = 10^{-4}$ consistently achieves near-optimal performance. Moreover, the optimal $\alpha$ decreases with more supervision, as expected, since the unsupervised regularization should receive less weight when paired data are abundant. Second, we evaluate the robustness of $\alpha$ when the unpaired data becomes increasingly out of distribution. We interpolate between clean unpaired data from CC3M and a fully OOD distribution by replacing a fraction $p$ of CC3M samples with random Gaussian noise, and then perform a grid search over $\alpha$ for each value of $p$. Figure 11b shows that SOTAlign remains robust across the full range of corruptions. The optimal value of $\alpha$ changes substantially only when the unpaired data are entirely OOD ($p = 1$), indicating graceful degradation rather than failure under distribution shift.

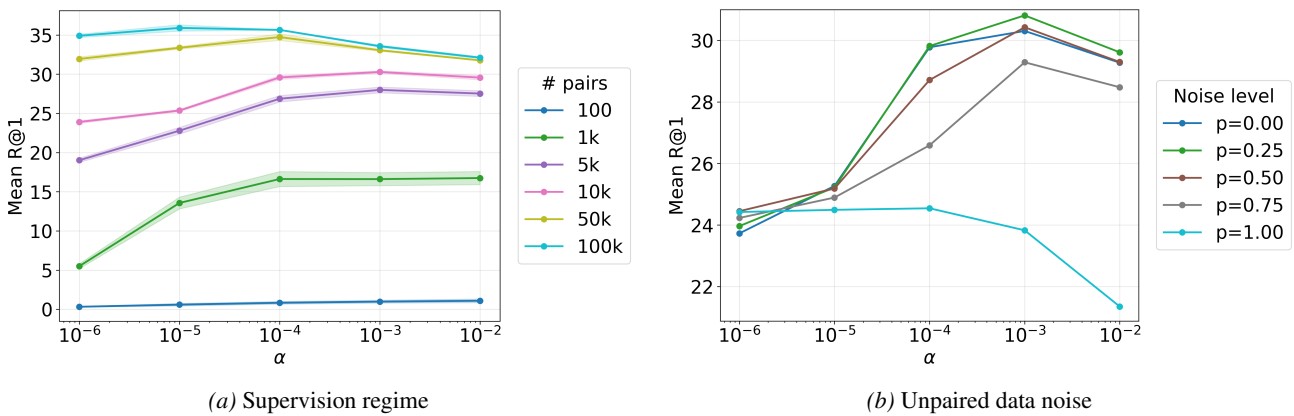

*(a)* Supervision regime            *(b)* Unpaired data noise

*Figure 11.* Sensitivity of SOTAlign to the regularization weight $\alpha$ under varying supervision regimes (number of supervised pairs) and varying noise level of the unpaired data.

*Table 8.* Robustness of SOTAlign to teacher noise. MeanR@1 on COCO under different teacher noise levels $p$ in comparison to the supervised baseline SAIL.

| Teacher Noise $p$ | 0.0 | 0.1 | 0.2 | 0.3 | 0.4 | 0.5 | SAIL |
|---|---|---|---|---|---|---|---|
| COCO MeanR@1 | 30.8 | 30.1 | 29.8 | 29.0 | 27.9 | 26.6 | 23.6 |

**OT Regularization.** We fix $\alpha = 0.001$, set the maximum number of Sinkhorn iterations to 500, and vary $\epsilon, \epsilon^* \in \{0.005, 0.01, 0.05, 0.1, 0.5\}$. Figure 12a and Figure 12b display this grid search for 1k and 10k supervised pairs, respectively. We recommend setting $\epsilon^* < \epsilon$ so that the teacher distribution has lower entropy than the student, encouraging the student to sharpen toward the teacher's geometry. SOTAlign is more sensitive to $\epsilon$ when supervision is scarce. Importantly, the same guidelines apply across supervision regimes and the optimal ratio $\epsilon/\epsilon^*$ remains consistent.

### B.3. Quantifying the Distribution Shift

We study the effect of the distribution shift arising from the use of unpaired data $(X, Y)$ drawn from different sources from those of the paired data $(A, B)$ using the total spherical sliced Wasserstein distance (see Appendix C.2) computed using the Python library POT (Flamary et al., 2021; 2024). In all experiments, we set $p = 2$ and use $N = 500$ projection directions. Distances between unimodal datasets are reported in Figure 15 averaged over 20 random seeds corresponding to a different subset of 100 000 samples of the dataset and projection set. All distances are computed between embeddings from Dinov3 and NV-Embed-V2 for images and text respectively.

In Figure 6 we exhibit a strong correlation between distance and performance, which provides a good proxy for performance that can be computed without any training or inference. In addition, this correlation is even stronger when one of the unimodal unpaired dataset is fixed to be CC3M and the other is varied. We show that performance is strongly correlated with the SSW distances between the unimodal datasets, $SSW(B, Y)$ and $SSW(A, X)$ in Figure 13 and 14 respectively.

### B.4. Benchmarking Semi-Supervised Alignment

Table 11 reports retrieval performance on COCO and Flickr30k, while Table 12 presents zero-shot image classification accuracy across a variety of downstream datasets.

In Section 6.5, we evaluate SOTAlign in a semi-supervised alignment setting proposed by Yacobi et al. (2025). We train for alignment on a single dataset, and evaluate retrieval on the test split of the same dataset (with only 400 test instances for retrieval). The datasets are: COCO (Lin et al., 2014), Flickr30 (Plummer et al., 2015), and Polyvore (Han et al., 2017). Yacobi et al. (2025) use MLPs for alignment and an embedding dimensionality of 8. In Table 13, we report the performance of SOTAlign adhering to their architectural choices. Our method achieves gains of +5.5 on COCO, +28.7 on Flickr30k, and +18.2 on Polyvore I2T R@5. However, if we lift these constraints and instead use linear alignment layers with a target dimension of 512, performance increases further, reaching +14.3 on COCO, +40.0 on Flickr30k, and +32.5 on Polyvore.

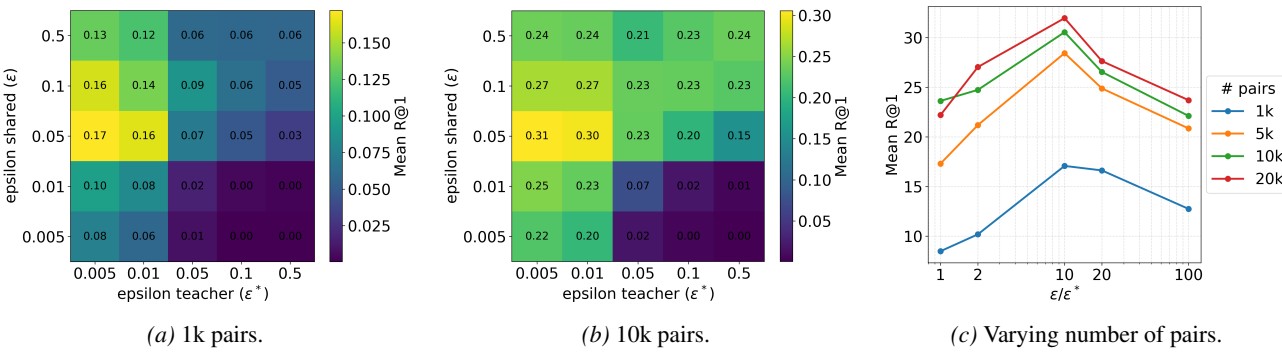

*(a)* 1k pairs.     *(b)* 10k pairs.     *(c)* Varying number of pairs.

*Figure 12.* OT regularization. We fix $\alpha = 0.001$, use 1M unpaired samples, and vary $\epsilon$ and $\epsilon^*$ across different supervision regimes.

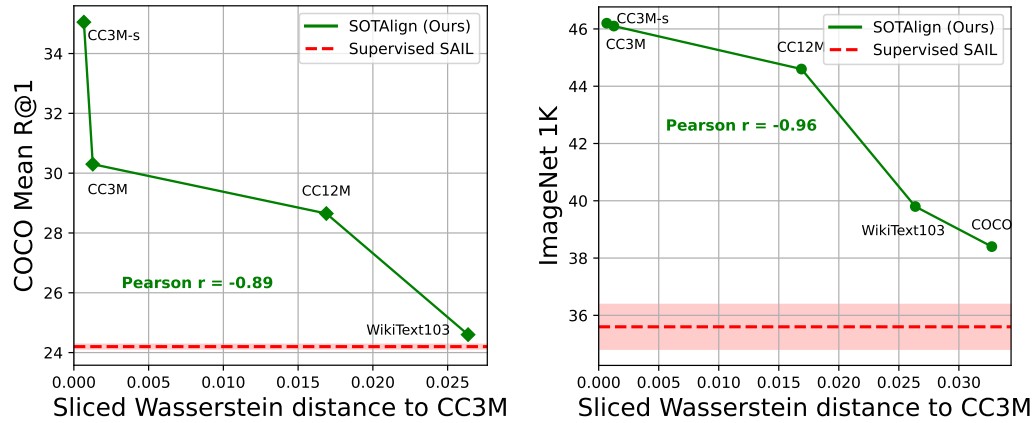

*Figure 13.* Performance when using CC3M as paired data, CC3M text as unpaired text, and other image datasets as unpaired images, together with a comparison to the spherical sliced Wasserstein distance between CC3M image and the other image datasets.

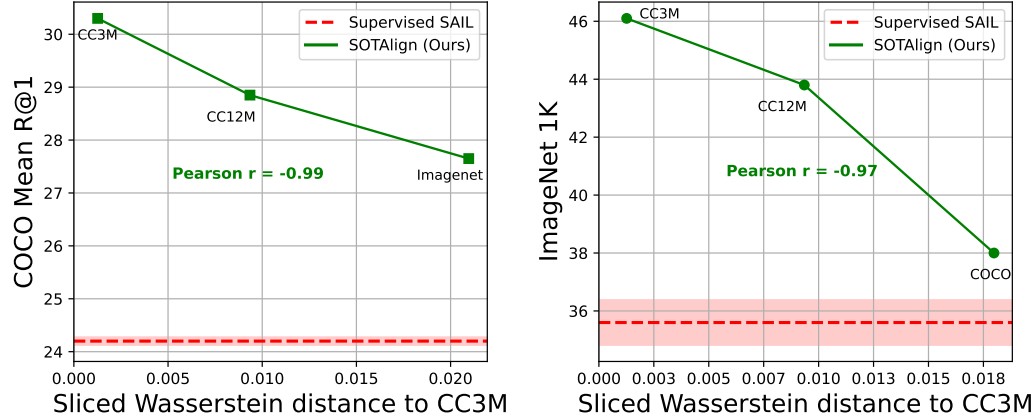

*Figure 14.* Performance when using CC3M as paired data, CC3M text as unpaired text, and other image datasets as unpaired images, together with a comparison to the spherical sliced Wasserstein distance between CC3M image and the other image datasets.

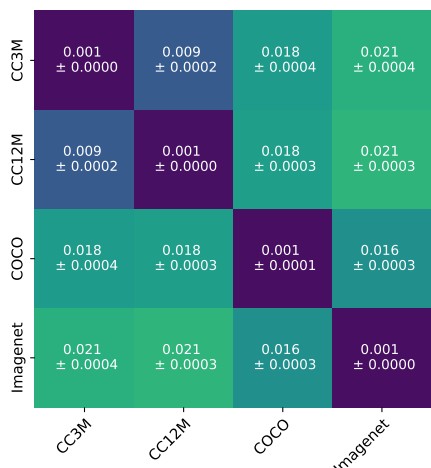 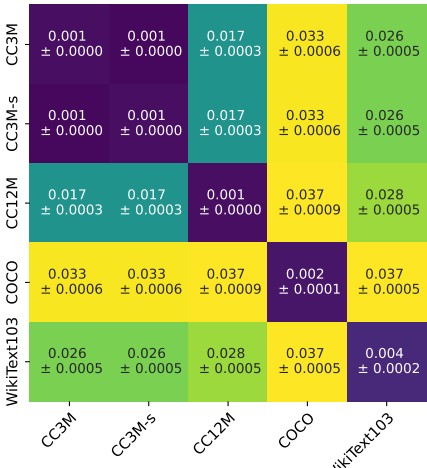

*Figure 15.* Spherical sliced Wasserstein distances between different image datasets (left) and text datasets (right). We report mean and std of the distances over 20 seeds.

As shown in Table 7, SOTAlign generalizes effectively to the domain-specific SkyScript setting, achieving strong mean R@10 performance when trained with 10k paired samples and 700k unpaired image and text samples. To provide a more comprehensive evaluation, Figure 16 and Figure 17 report retrieval results across different supervision regimes and evaluation metrics, including R@1, R@5, and R@10. These figures further compare performance under different test set sizes with 10k and 30k SkyScript samples, respectively.

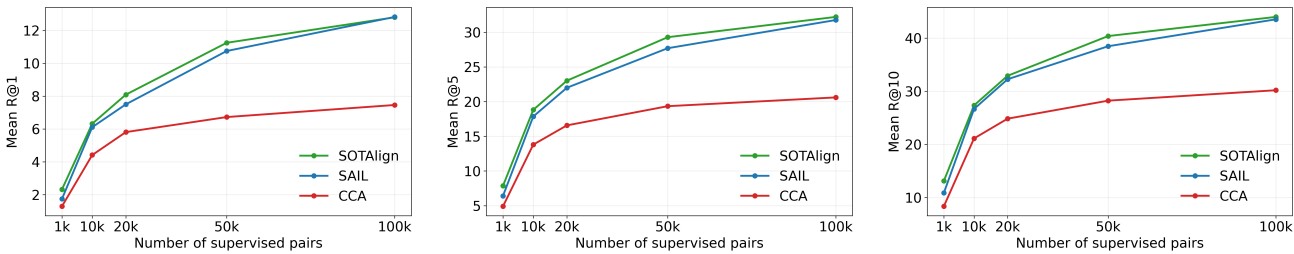

*Figure 16.* Satellite image-text retrieval on SkyScript with 10k test pairs.

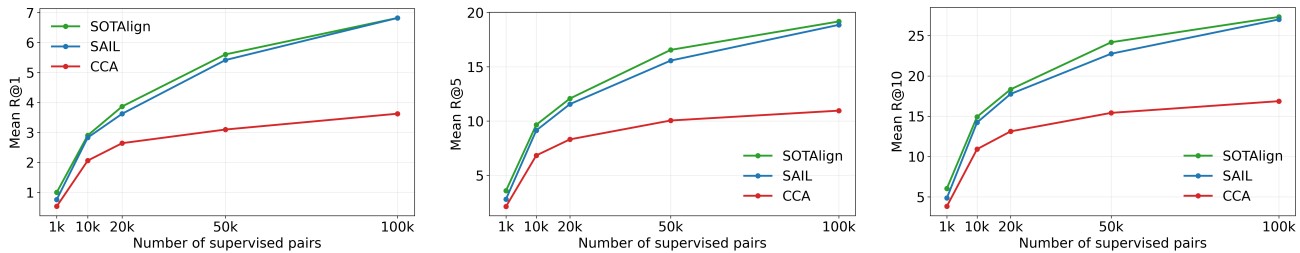

*Figure 17.* Satellite image-text retrieval on SkyScript with 30k test pairs.

*Table 9.* We train SOTAlign on 10k image-text pairs from CC3M and up to 1M samples from varying unimodal datasets and report the zero-shot classification (top-1 accuracy) and retrieval (R@1). For comparison we report SAIL trained with as many samples and SAIL 1M i.e. the version trained using 1M supervised samples from CC3M (100x more supervision).

| Method | Unpaired Images | Unpaired Text | ImageNet-1K | COCO T2I | COCO I2T |
|---|---|---|---|---|---|
| SAIL 1M | — | — | 56.4 | 35.5 | 45.5 |
| SAIL | — | — | 35.6 | 21.0 | 27.4 |
| SOTAlign | CC3M | CC3M | 46.1 | 26.5 | 34.1 |
| SOTAlign | CC3M | CC3M synth. | 46.2 | **30.4** | **39.7** |
| SOTAlign | CC3M | CC12M | 44.6 | 24.5 | 32.0 |
| SOTAlign | CC12M | CC3M | 43.8 | 24.9 | 32.8 |
| SOTAlign | CC12M | CC12M | **46.8** | 25.7 | 34.4 |
| SOTAlign | ImageNet | CC3M | 43.4 | 23.8 | 31.5 |
| SOTAlign | ImageNet | CC12M | 44.3 | 24.2 | 31.5 |
| SOTAlign | CC3M | COCO | 38.4 | 28.4 | 26.7 |
| SOTAlign | CC12M | COCO | 38.3 | 27.7 | 30.6 |
| SOTAlign | ImageNet | COCO | 40.6 | 25.5 | 26.1 |
| SOTAlign | COCO | CC3M | 38.0 | 21.7 | 33.9 |
| SOTAlign | COCO | CC12M | 38.5 | 22.0 | 33.1 |
| SOTAlign | CC3M | WikiText103 | 39.8 | 21.4 | 27.8 |
| SOTAlign | COCO | WikiText103 | 37.1 | 19.5 | 29.4 |
| SOTAlign | ImageNet | WikiText103 | 40.7 | 20.7 | 28.1 |

*Table 10.* SOTAlign trained on 10k pairs and 1M unpaired samples from CC3M with different unimodal encoders. Zero-shot classification (top-1 accuracy) and retrieval (R@1). Green values represent the absolute gain over supervised SAIL.

| Vision Model | Language Model | Mutual k-NN | Method | ImageNet 1K | COCO T2I | COCO I2T |
|---|---|---|---|---|---|---|
| DINOv2 | Nemotron-8B | 14.6 | SAIL | 25.5 | 11.7 | 18.0 |
| | | | SOTAlign | 32.4 +6.9 | 15.5 +3.8 | 23.3 +5.3 |
| | Qwen3-8B | 18.9 | SAIL | 31.8 | 16.8 | 23.8 |
| | | | SOTAlign | 39.5 +7.7 | 20.9 +4.1 | 31.1 +7.3 |
| | NV-Embed-v2 | 18.2 | SAIL | 32.7 | 19.0 | 23.4 |
| | | | SOTAlign | 42.5 +9.8 | 23.1 +4.1 | 31.1 +7.7 |
| DINOv3 | Nemotron-8B | 14.1 | SAIL | 25.4 | 12.8 | 21.0 |
| | | | SOTAlign | 35.5 +10.1 | 16.6 +3.8 | 26.2 +5.2 |
| | Qwen3-8B | 18.0 | SAIL | 33.4 | 19.1 | 28.0 |
| | | | SOTAlign | 42.7 +9.3 | 24.1 +5.0 | **35.3** +7.3 |
| | NV-Embed-v2 | 17.6 | SAIL | 35.6 | 21.0 | 27.4 |
| | | | SOTAlign | **46.1** +10.5 | **26.5** +5.5 | 34.1 +6.7 |

*Table 11.* Zero-shot text-image retrieval (Recall@K) on COCO and Flickr30k. Comparison of supervised and semi-supervised methods trained with 10k image-text pairs and 1M unpaired samples from CC3M. Upper bound with 1M supervised pairs in grey.

|  | Method | COCO | | | | Flickr30k | | | |
|---|---|---|---|---|---|---|---|---|---|
|  |  | T2I | | I2T | | T2I | | I2T | |
|  |  | R@1 | R@5 | R@1 | R@5 | R@1 | R@5 | R@1 | R@5 |
| Sup. | SAIL (1M) | 35.5 | 61.0 | 45.5 | 72.0 | 63.1 | 87.2 | 75.0 | 94.4 |
| | SAIL | 21.0 | 44.3 | 27.4 | 51.7 | 45.7 | 75.1 | 54.1 | 81.2 |
| | STRUCTURE | 21.0 | 43.7 | 28.7 | 52.7 | 46.8 | 74.9 | 54.0 | 82.9 |
| Semi-sup. | SAIL | 20.7 | 43.6 | 26.5 | 51.7 | 44.9 | 74.2 | 53.1 | 82.1 |
| | STRUCTURE | 20.9 | 43.5 | 28.0 | 52.2 | 45.7 | 74.9 | 56.0 | 83.3 |
| | NNCLR | 21.3 | 44.4 | 27.9 | 52.2 | 46.6 | 75.3 | 52.9 | 82.1 |
| | S-CLIP | 20.4 | 42.6 | 27.8 | 50.5 | 44.5 | 74.4 | 52.6 | 82.3 |
| | **SOTAlign (Ours)** | **26.5** | **49.8** | **34.1** | **59.4** | **51.7** | **79.2** | **60.8** | **85.7** |

*Table 12.* Zero-shot image classification (top-1 accuracy). Comparison of supervised and semi-supervised methods trained with 10k image-text pairs and 1M unpaired samples from CC3M. Upper bound with 1M supervised pairs in grey..

|  | Method | Food-101 | CIFAR-10 | CIFAR-100 | Aircraft | DTD | Flowers | ImageNet |
|---|---|---|---|---|---|---|---|---|
| Sup. | SAIL (1M) | 63.9 | 97.8 | 82.3 | 9.7 | 53.5 | 47.2 | 56.4 |
| | SAIL | 36.4 | 96.2 | 71.2 | 3.9 | 36.8 | 24.1 | 35.6 |
| | STRUCTURE | 38.5 | 96.7 | 72.2 | **5.4** | 39.5 | 23.6 | 38.2 |
| Semi-sup. | SAIL | 36.5 | 96.2 | 71.3 | 3.8 | 35.9 | 21.1 | 35.6 |
| | STRUCTURE | 37.6 | 96.5 | 70.8 | 4.9 | 38.7 | 23.2 | 36.8 |
| | NNCLR | 37.9 | 96.5 | 73.0 | 3.8 | 38.8 | 24.6 | 37.4 |
| | S-CLIP | 35.3 | 95.9 | 69.3 | 4.4 | 37.6 | 22.5 | 36.4 |
| | **SOTAlign (Ours)** | **50.0** | **97.5** | **78.3** | 5.0 | **42.4** | **30.1** | **46.1** |

*Table 13.* Alignment per dataset. Following the setup of SUE (Yacobi et al., 2025), we train for alignment per dataset and evaluate image-text retrieval (Recall@5). We report SOTAlign results adhering to the architectural choices of SUE (MLP, embedding dimensionality of 8) and without these constraints.

| | COCO 100 Pairs | | Flickr30k 500 Pairs | | Polyvore 500 Pairs | |
|---|---|---|---|---|---|---|
| **Method** | I2T | T2I | I2T | T2I | I2T | T2I |
| CSA | 1.3 | 1.0 | 1.3 | 0.8 | 1.3 | 1.0 |
| Contrastive | 8.5 | 5.8 | 9.5 | 9.8 | 13.8 | 11.5 |
| SUE | 21.5 | 18.3 | 19.8 | 22.0 | 22.8 | 20.8 |
| **SOTAlign** (with SUE constraints) | 27.0 | 28.8 | 48.5 | 48.8 | 41.0 | 39.8 |
| **SOTAlign** (without SUE constraints) | **35.8** | **35.0** | **59.8** | **63.3** | **55.3** | **55.3** |

# C. Mathematical Details

## C.1. Linear Alignment Models

We now provide the closed-form solutions for the proposed linear alignment models.

**Procrustes.** The classical Orthogonal Procrustes problem is defined for two point clouds $A, B \in \mathbb{R}^{n \times d}$ and seeks an orthogonal transformation that best aligns $A$ to $B$. It can be written as

$$\max_{P \in \mathbb{R}^{d \times d}} \langle PA^\top, B \rangle \quad \text{s.t. } PP^\top = I_d. \tag{16}$$

This formulation learns a single linear mapping from $A$ to $B$ and implicitly assumes that both point clouds lie in the same ambient space $\mathbb{R}^d$.

However, Procrustes alignment is known to admit flexible generalizations beyond this setting (Gower & Dijksterhuis, 2004). In particular, it can be extended to handle representations of different dimensionalities and to learn projections into a shared lower-dimensional space. We now introduce a natural variant of Procrustes alignment that is better suited to our setting.

**Proposition C.1** (Closed form solution of Procrustes Alignment). *Let $A \in \mathbb{R}^{n \times d_a}$ and $B \in \mathbb{R}^{n \times d_b}$, and let $d' \leq \min\{d_a, d_b\}$. Consider the optimization problem*

$$\max_{P \in \mathbb{R}^{d' \times d_a}, Q \in \mathbb{R}^{d' \times d_b}} \langle PA^\top, BQ^\top \rangle \quad \text{s.t. } PP^\top = I_{d'}, QQ^\top = I_{d'}. \tag{17}$$

*Let the singular value decomposition of $A^\top B$ be*

$$A^\top B = U\Sigma V^\top,$$

*with singular values in non-increasing order. Then an optimal solution is given by*

$$W_x = U_{:,1:d'}, \qquad W_y = V_{:,1:d'}.$$

*Proof.* We introduce the change of variables

$$\tilde{P} = PU, \qquad \tilde{Q} = QV.$$

Since $U$ and $V$ are orthogonal, $\tilde{P}$ and $\tilde{Q}$ also satisfy $\tilde{P}\tilde{P}^\top = \tilde{Q}\tilde{Q}^\top = I_{d'}$. Using invariance of the Frobenius inner product under orthogonal transformations, the objective rewrites as

$$\langle PA^\top, BQ^\top \rangle = \langle \tilde{P}\Sigma, \tilde{Q} \rangle.$$

By the Cauchy–Schwarz inequality,

$$\langle \tilde{P}\Sigma, \tilde{Q} \rangle \leq \|\tilde{P}\Sigma\|_F \|\tilde{Q}\|_F.$$

Since $\tilde{Q} \in \mathbb{R}^{d' \times d_b}$ has orthonormal rows, we have

$$\|\tilde{Q}\|_F^2 = \text{tr}(\tilde{Q}\tilde{Q}^\top) = d'.$$

We now bound $\|\tilde{P}\Sigma\|_F^2$. By definition,

$$\|\tilde{P}\Sigma\|_F^2 = \text{tr}(\tilde{P}\Sigma^2\tilde{P}^\top) = \text{tr}(\Sigma^2\tilde{P}^\top\tilde{P}),$$

where we used cyclic invariance of the trace. Since $\tilde{P}$ has orthonormal rows, the matrix

$$\Pi = \tilde{P}^\top\tilde{P}$$

is an orthogonal projector of rank $d'$, with eigenvalues in $\{0, 1\}$ and $\text{tr}(\Pi) = d'$.

Let $\Sigma^2 = \text{diag}(\sigma_1^2, \ldots, \sigma_r^2)$ with $\sigma_1 \geq \sigma_2 \geq \cdots \geq \sigma_r \geq 0$. Then

$$\text{tr}(\Sigma^2 \Pi) = \sum_{i=1}^{r} \sigma_i^2 \, \Pi_{ii}.$$

Because $0 \leq \Pi_{ii} \leq 1$ for all $i$ and $\sum_i \Pi_{ii} = d'$, the sum is maximized by assigning weight 1 to the $d'$ largest diagonal entries of $\Sigma^2$. Therefore,

$$\text{tr}(\Sigma^2 \Pi) \leq \sum_{i=1}^{d'} \sigma_i^2.$$

Combining the above bounds yields

$$\langle \tilde{P}\Sigma, \tilde{Q} \rangle \leq \sqrt{d'} \left( \sum_{i=1}^{d'} \sigma_i^2 \right)^{1/2},$$

and the bound is tight when $\tilde{P} = \tilde{Q} = I_{d'}$ which concludes the proof. $\qquad\square$

**Canonical Correlation Analysis (CCA).**  Canonical Correlation Analysis (CCA) is a classical tool for studying linear relationships between two sets of variables and is widely used in multivariate statistics and representation learning. In this work, CCA is already defined in (7); we briefly recall its formulation here in a form that is convenient for deriving its closed-form solution and for highlighting its connection to Procrustes alignment.

Denoting

$$\Sigma_{x,x} = A^\top A, \qquad \Sigma_{x,y} = A^\top B, \qquad \Sigma_{y,y} = B^\top B,$$

the CCA problem (7) can be equivalently rewritten as

$$(W_x, W_y) = \arg\max_{P,Q} \, \langle P\Sigma_{x,y}, Q \rangle$$
$$\text{s.t.} \quad P\Sigma_{x,x}P^\top = I_{d'}, \qquad Q\Sigma_{y,y}Q^\top = I_{d'}. \tag{18}$$

We now present a standard derivation of the closed-form solution, included for completeness, which makes explicit the relationship between CCA and the Procrustes problem introduced above.

**Proposition C.2** (Closed-form solution of CCA). *Let*

$$\Sigma_{x,x}^{-1/2} \Sigma_{x,y} \Sigma_{y,y}^{-1/2} = U\Sigma V^\top$$

*be the singular value decomposition, with singular values in non-increasing order. Then an optimal solution to* (18) *is given by*

$$W_x = U_{:,1:d'}^\top \Sigma_{x,x}^{-1/2}, \qquad W_y = V_{:,1:d'}^\top \Sigma_{y,y}^{-1/2}.$$

*Proof.* We introduce the change of variables

$$\tilde{P} = P\Sigma_{x,x}^{1/2}, \qquad \tilde{Q} = Q\Sigma_{y,y}^{1/2}.$$

Under this transformation, the constraints become

$$\tilde{P}\tilde{P}^\top = I_{d'}, \qquad \tilde{Q}\tilde{Q}^\top = I_{d'},$$

and the objective rewrites as

$$\langle P\Sigma_{x,y}, Q \rangle = \left\langle \tilde{P} \, \Sigma_{x,x}^{-1/2}\Sigma_{x,y}\Sigma_{y,y}^{-1/2}, \tilde{Q} \right\rangle.$$

Thus, the CCA problem reduces to an orthogonal Procrustes problem:

$$\max_{\tilde{P}\tilde{P}^\top = \tilde{Q}\tilde{Q}^\top = I_{d'}} \left\langle \tilde{P}M, \tilde{Q} \right\rangle, \quad \text{where } M = \Sigma_{x,x}^{-1/2}\Sigma_{x,y}\Sigma_{y,y}^{-1/2}.$$

Let $M = U\Sigma V^\top$ be its singular value decomposition. By the Procrustes result, the maximum is attained for

$$\tilde{P} = U_{:,1:d'}^\top, \qquad \tilde{Q} = V_{:,1:d'}^\top.$$

Substituting back yields

$$P = U_{:,1:d'}^\top \Sigma_{x,x}^{-1/2}, \qquad Q = V_{:,1:d'}^\top \Sigma_{y,y}^{-1/2},$$

which concludes the proof. $\qquad\square$

### C.2. Optimal Transport

**Introduction to Optimal Transport**   We briefly recall the discrete optimal transport (OT) problem and its entropic relaxation. We refer to (Peyré & Cuturi, 2019) for more details. Let $n \in \mathbb{N}$ and denote by $\mathcal{P}_n$ the set of permutation matrices,

$$\mathcal{P}_n = \{P \in \{0,1\}^{n \times n} \mid P\mathbf{1} = \mathbf{1}, \ P^\top \mathbf{1} = \mathbf{1}\}, \qquad \text{(Permutations Matrices)}$$

and by $\Pi_n$ the set of bistochastic matrices,

$$\Pi_n = \{T \in \mathbb{R}_+^{n \times n} \mid T\mathbf{1} = \mathbf{1}, \ T^\top \mathbf{1} = \mathbf{1}\}. \qquad \text{(Transport Plans)}$$

We further define the (negative) entropy of a transport plan $T \in \Pi_n$ as

$$H(T) = \sum_{i,j} T_{ij} \log T_{ij}. \qquad \text{(Negative Entropy)}$$

In the discrete OT setting, we are given two sets of points indexed by $i, j \in \{1, \ldots, n\}$ and a cost matrix $C \in \mathbb{R}^{n \times n}$, where $C_{ij}$ denotes the cost of transporting mass from point $i$ to point $j$. The classical Monge formulation seeks the permutatin minimizing the total transport cost,

$$\min_{P \in \mathcal{P}_n} \langle P, C \rangle. \qquad \text{(Monge)}$$

This formulation enforces a one-to-one matching and is combinatorial in nature.

Kantorovich proposed a convex relaxation of this problem by allowing fractional transport plans,

$$\min_{T \in \Pi_n} \langle T, C \rangle, \qquad \text{(Kantorovich)}$$

which can be shown to be equivalent to the Monge formulation in the discrete balanced setting (Peyré & Cuturi, 2019), while being more flexible and amenable to generalizations such as non-uniform marginals and continuous measures.

When the cost matrix is defined as $C_{i,j} = d(x_i, y_j)^p$, where $d$ is a distance on the underlying space and $p \geq 1$, the optimal value of the Kantorovich problem induces the p-Wasserstein distance, defined as

$$W_p = \left( \min_{T \in \Pi_n} \langle T, C \rangle \right)^{1/p}. \qquad \text{(Wasserstein distance)}$$

To further improve computational tractability, Cuturi (2013) introduced the entropic regularized OT problem, also known as the Sinkhorn relaxation,

$$\min_{T \in \Pi_n} \ \langle T, C \rangle + \varepsilon H(T), \qquad (19)$$

where $\varepsilon > 0$ controls the strength of the regularization. This formulation yields a strictly convex objective and can be efficiently solved using the Sinkhorn algorithm.

**Sliced Wasserstein distance**   Although entropically regularized optimal transport can be efficiently solved using the Sinkhorn algorithm, its computational complexity remains $O(n^2)$, which becomes prohibitive when comparing distributions supported on millions of high-dimensional points. To address this limitation, the sliced Wasserstein distance (SW) was introduced (Bonneel et al., 2015). The key observation underlying this approach is that the Wasserstein distance between one-dimensional distributions admits a closed-form solution obtained by sorting the samples and matching them

monotonically. The sliced Wasserstein distance exploits this property by projecting high-dimensional distributions onto multiple one-dimensional subspaces and averaging the resulting one-dimensional Wasserstein distances.

Let $\mu = \sum_{i=1}^{n} a_i \delta_{x_i}$ and $\nu = \sum_{j=1}^{m} b_i \delta_{y_j}$ be two discrete probability measures with $x_i, y_j \in \mathbb{R}^d$. For a given projection $\theta \in \mathbb{S}^{d-1}$, we define the projected one dimensional distributions as

$$\mu^{\theta} = \sum_{i=1}^{n} a_i \delta_{\langle x_i, \theta \rangle}, \quad \nu^{\theta} = \sum_{j=1}^{m} b_j \delta_{\langle y_j, \theta \rangle}. \tag{20}$$

Given a set of projection directions $(\theta_1, ..., \theta_N)$, the p-SW distance is defined as

$$SW_p(\mu, \nu) = \left( \frac{1}{N} \sum_{i=1}^{N} W_p^p(\mu^{\theta_i}, \nu^{\theta_i}) \right)^{1/p}. \tag{21}$$

When data are constrained to the unit sphere, the spherical sliced Wasserstein (SSW) distance (Liu et al., 2024) replaces linear projections with angular projections and computes optimal transport on the circle, thereby respecting the intrinsic geometry of directional data.

**Total sliced Wasserstein distance**   We introduce the total spherical sliced Wasserstein distance d as a measure of the distribution shift between a source of unpaired data $\mathcal{D} = (X, Y)$ and the paired dataset $\mathcal{D}_p = (A, B)$. Using the spherical sliced Wasserstein distance in place of the standard sliced Wasserstein distance ensures that the resulting distances between text distributions and between image distributions are computed on a comparable scale, enabling fair comparison across modalities. The total SSW distance is defined as

$$d(\mathcal{D}, \mathcal{D}_p) = \text{SSW}(X, A) + \text{SSW}(Y, B). \tag{22}$$

**Theoretical Results.**   We now present the theoretical contribution underlying our proposed divergence and its efficient differentiation. Throughout, we work with an affinity matrix $K \in \mathbb{R}^{n \times n}$ rather than a cost matrix, following the convention $C = -K$ for consistency with the rest of the paper.

For any transport plan $T \in \Pi_n$, we define the entropic OT objective

$$W_\epsilon(T, K) = -\langle T, K \rangle + \epsilon H(T), \tag{23}$$

and the associated optimal value

$$W_\epsilon(K) = \min_{T \in \Pi_n} W_\epsilon(T, K). \tag{24}$$

Finally, we denote

$$\text{OT}_\epsilon(K) = \underset{T \in \Pi_n}{\arg \min} \, W_\epsilon(T, K) \tag{25}$$

the corresponding optimal transport plan.

Importantly, we recall the following fundamental result in entropic optimal transport states that there exist dual potentials $u, v \in \mathbb{R}^n$ such that the optimal transport plan admits the decomposition

$$\log \text{OT}_\epsilon(K) = u\mathbf{1}^\top + \frac{K}{\epsilon} + \mathbf{1}v^\top, \tag{26}$$

see e.g. Peyré & Cuturi (2019). This characterization allows us to establish the following lemma.

**Lemma C.3.** *For any transport plan* $T \in \Pi_n$,

$$\langle T, \log OT_\epsilon(K) \rangle = \frac{\langle T, K \rangle + W_\epsilon(K)}{\epsilon}. \tag{27}$$

*Proof.* For any $T \in \Pi_n$, we have

$$\langle T, \log \text{OT}_\epsilon(K) \rangle = \langle T, u\mathbf{1}^\top \rangle + \langle T, \mathbf{1}v^\top \rangle + \frac{1}{\epsilon} \langle T, K \rangle.$$

Since $T$ is bistochastic, $\langle T, u\mathbf{1}^\top \rangle = \langle \mathbf{1}, u \rangle$ and $\langle T, \mathbf{1}v^\top \rangle = \langle \mathbf{1}, v \rangle$, yielding

$$\langle T, \log \text{OT}_\epsilon(K) \rangle = \langle \mathbf{1}, u + v \rangle + \frac{1}{\epsilon} \langle T, K \rangle.$$

In particular, setting $T = \text{OT}_\epsilon(K)$ yields

$$H(\text{OT}_\epsilon(K)) = \langle \mathbf{1}, u + v \rangle + \frac{1}{\epsilon} \langle \text{OT}_\epsilon(K), K \rangle.$$

which recovers a classical duality result

$$\langle \mathbf{1}, u + v \rangle = \frac{W_\epsilon(K)}{\epsilon}.$$

which gives the result. $\qquad\square$

Our main theoretical result follows by combining Lemma C.3 with the envelope theorem, yielding an explicit expression for the gradient of the proposed divergence.

**Theorem C.4.** *For any transport plan $T \in \Pi_n$,*

$$\nabla_K \text{KL}(T \| \text{OT}_\epsilon(K)) = \frac{\text{OT}_\epsilon(K) - T}{\epsilon}. \tag{28}$$

*Proof.* By definition,

$$\begin{aligned} \text{KL}(T \| \text{OT}_\epsilon(K)) &= \left\langle T, \log \frac{T}{\text{OT}_\epsilon(K)} \right\rangle \\ &= \langle T, \log T \rangle - \langle T, \log \text{OT}_\epsilon(K) \rangle \end{aligned}$$

and only the second term depends on $K$. Differentiating yields

$$\nabla_K \text{KL}(T \| \text{OT}_\epsilon(K)) = -\nabla_K \langle T, \log \text{OT}_\epsilon(K) \rangle.$$

Applying Lemma C.3 gives

$$\nabla_K \text{KL}(T \| \text{OT}_\epsilon(K)) = -\frac{1}{\epsilon} \nabla_K \big( \langle T, K \rangle + W_\epsilon(K) \big).$$

Since $W_\epsilon(K)$ is defined as the minimum of $W_\epsilon(T, K)$ over $T \in \Pi_n$ which is a strongly convex problem, the envelope theorem implies

$$\nabla_K W_\epsilon(K) = \nabla_K W_\epsilon(\text{OT}_\epsilon(K), K) = -\text{OT}_\epsilon(K)$$

from which the result follows. $\qquad\square$

### C.3. Centered Kernel Alignment (CKA)

Centered Kernel Alignment (CKA) (Cristianini et al., 2001) is a widely used measure of similarity between representation spaces, defined in terms of their associated kernel (or Gram) matrices. Let $H = I_n - \frac{1}{n}\mathbf{1}\mathbf{1}^\top$ denote the centering matrix and $\| \cdot \|_F$ the Frobenius norm. Given two kernel matrices $K_1, K_2 \in \mathbb{R}^{n \times n}$, CKA is defined as

$$\text{CKA}(K_1, K_2) = \frac{\langle K_1 H, H K_2 \rangle}{\sqrt{\langle K_1 H, H K_1 \rangle \langle K_2 H, H K_2 \rangle}}. \tag{29}$$

For the sake of completeness we now share a few classical results regarding CKA.

**Kernel centering.** The matrix $H$ plays the role of centering the data in feature space. We define the *centered kernel* as

$$\bar{K} = HKH. \tag{30}$$

This operation corresponds to centering the underlying representations before computing pairwise similarities. Indeed, in the linear case where $K = XX^\top$ for data matrix $X \in \mathbb{R}^{n \times d}$, we have

$$\bar{K} = \bar{X}\bar{X}^\top, \tag{31}$$

where $\bar{X}$ denotes the centered features $\bar{X}_i = X_i - \frac{1}{n}\sum_{j=1}^n X_j$, i.e., $\bar{X} = HX$.

**CKA as a cosine affinity.** A well known property of CKA is that it can be interpreted as a cosine similarity between centered kernels, viewed as vectors in $\mathbb{R}^{n^2}$.

**Proposition C.5.** *Let $\bar{K}_1 = HK_1H$ and $\bar{K}_2 = HK_2H$. Then CKA can be written as*

$$\mathrm{CKA}(K_1, K_2) = k\big(\mathrm{vec}(\bar{K}_1), \mathrm{vec}(\bar{K}_2)\big), \tag{32}$$

*where $k(\cdot, \cdot)$ denotes the cosine affinity and $\mathrm{vec}(\cdot)$ denotes matrix vectorization.*

*Proof.* Recall that $H = I_n - \frac{1}{n}\mathbf{1}\mathbf{1}^\top$ is symmetric and idempotent, i.e., $H^\top = H$ and $H^2 = H$. We compute

$$\begin{aligned}
\langle HK_1H, HK_2H \rangle &= \mathrm{tr}(HK_1H\,HK_2H) \\
&= \mathrm{tr}(HK_1HK_2H) \\
&= \mathrm{tr}(K_1HK_2H) \\
&= \langle K_1H, HK_2 \rangle,
\end{aligned}$$

where we used cyclic invariance of the trace and the idempotence of $H$.

In particular, setting $K_1 = K_2 = K$ yields

$$\langle HKH, HKH \rangle = \|HKH\|_F^2. \tag{33}$$

Combining these identities proves that CKA is exactly the cosine similarity between the vectorized centered kernels. $\square$

**Computational Complexity.** We conclude this section by providing the computational complexity of CKA for linear kernels, as considered in this work.

**Proposition C.6.** *Assume that*

$$K_1 = X_1X_1^\top \quad \text{with } X_1 \in \mathbb{R}^{n \times d_1}, \qquad K_2 = X_2X_2^\top \quad \text{with } X_2 \in \mathbb{R}^{n \times d_2},$$

*and denote $d = \max(d_1, d_2)$. Then the memory complexity of computing $\mathrm{CKA}(K_1, K_2)$ is*

$$\mathcal{O}\big(nd + d^2\big).$$

*Proof.* Assume that $X_1$ and $X_2$ are centered, which can be done in $\mathcal{O}(nD)$ time and memory. Using the identities established above, we have

$$\langle K_1H, HK_2 \rangle = \langle X_1X_1^\top, X_2X_2^\top \rangle = \langle X_1^\top X_2, X_1^\top X_2 \rangle = \|X_1^\top X_2\|_F^2.$$

Thus, computing the numerator only requires storing the $d_1 \times d_2$ matrix $X_1^\top X_2$.

Similarly,

$$\|K_1H\|_F^2 = \|X_1^\top X_1\|_F^2, \qquad \|K_2H\|_F^2 = \|X_2^\top X_2\|_F^2,$$

which require storing only the $d_1 \times d_1$ and $d_2 \times d_2$ Gram matrices, respectively.

Therefore, the overall memory complexity is dominated by storing $X_1, X_2$ and the associated Gram matrices, yielding

$$\mathcal{O}(nD + d^2),$$

as claimed. $\square$

