# OpenReview forum: "SOTAlign: Semi-Supervised Alignment of Unimodal Vision and Language Models via Optimal Transport"
_ICML.cc/2026/Conference — ICML 2026 regular_

### Official Review · Reviewer_1eos · 2026-03-11

**Soundness:** 3
**Presentation:** 3
**Significance:** 3
**Originality:** 3
**Overall Recommendation:** 4
**Confidence:** 3

**Summary:**

The paper studies how to align frozen pretrained vision and language with a small number of image-text pairs models using lightweight alignment layers. Traditional vision-language models like CLIP rely on contrastive loss and millions of paired samples. This makes VLMs costly to train and difficult to adapt when paired data annotations are limited. In this work the authors have investigated vision-language alignment beyond large-scale supervision. Thereby discussing whether meaningful cross-modal alignment can be achieved from pretrained encoders using limited paired supervision and large amounts of unpaired text and data providing empirical support for the Platonic Representation Hypothesis.

The paper introduces SOTAlign, a semi-supervised framework for aligning frozen unimodal encoders. The first stage is a simple linear alignment model with the supervised pairs (A,B) to capture a coarse shared structure between modalities. Secondly, the model relies on a KLOT, an optimal-transport based objective that refines this alignment using large amounts of unpaired data by minimizing the divergence between affinity matrices derived from the embeddings. They derive explicit gradient of KLOT to avoid the computational bottlenecks associated with optimal transport gradients. Extensive experiments against supervised and semi-supervised baselines shows improved performance on image-text retrieval task and zero-shot classification.

**Compliance With Llm Reviewing Policy:**

Affirmed.

**Final Justification:**

The authors have adequately addressed the reviewers’ comments and clarified several points in the revised manuscript. However, while these changes improve readability and presentation, they do not alter the core contributions or overall impact of the work. Therefore, my evaluation and score remain unchanged.

**Key Questions For Authors:**

1.	**Sensitivity to the Linear Teacher:** How sensitive is the SOTAlign to the quality of the initial linear alignment method? Will the model refinement degrade with poor teacher alignment?
2.	**Robustness to Domain Shift:** Could the authors provide more analysis on how severe domain shifts affect the effectiveness of the OT-based regularization?
3.	**Computational Cost:** Although the explicit gradient reduces memory usage, what is the practical runtime overhead compared to contrastive training methods?

**Limitations:**

Partially the authors have briefly mentioned societal impact but does not discuss limitations. It would be good if authors included discussion on limitations like: dependence on pretrained encoders and possible performance degradation under extreme distribution shifts.

**Strengths And Weaknesses:**

**Soundness:** The work is very technical, and the problem formulation is clearly defined. The two-stage design (linear teacher + refinement) is a reasonable solution and aligns with their hypothesis (Platonic Representation) that *“pretrained unimodal image and text encoders may already produce semantically aligned representations, even in the absence of explicit cross-modal training.”* The KLOT divergence with explicit gradient is a good extension of optimal transport-based methods. This avoids the memory and computation overhead of Sinkhorn unrolling by 50x.

The authors have fairly extensive experiments and an ablation study: Paired samples, Unpaired samples, Divergence choices, Vision-language encoders, etc. All these results show consistent improvement over baseline in both retrieval and classification tasks.

**Weakness:**
1. I would request the author to have a discussion on statistical variance to strengthen their claim.
2. Some comparisons rely on adapted baselines rather than widely established semi-supervised alignment methods making it difficult to compare them relative to SOTA works.
3. Missing comparisons with existing OT-based models like OT-CLIP, UOT-RCL.

**Presentation:** The paper is well written and easy to follow. Motivation followed by Problem formulation with Method description and Experimental results. The figures are also self-explanatory on the pipeline. Overall, the presentation quality is good but can be improved with clearer explanations in the methodological sections.

**Suggestions:**
1. Some portions on the OT part are dense with equations making it little difficult to follow.
2. The discussion of related works could highlight how the proposed approach differs from prior OT-based alignment methods.

**Significance:** The problem described in the paper is crucial. Large-scale paired annotation is expensive in many domains making their model suitable practical approach in these medical imaging or scientific data domains.
The results show that SOTAlign can significantly reduce the amount of required supervision while maintaining strong performance marking its impact in vision-language models. This motivates future directions on multimodal representation learning and semi-supervised alignment.

**Originality:** The work contains several elements: Linear teacher alignment, Semi-supervised refinement, and new optimal transport divergence (KLOT) with efficient gradient formulation. While optimal transport has been explored before in representation alignment, the formulation of KLOT for semi-supervised vision-language alignment appears to be new. They empirically prove the hypothesis. Overall, the work demonstrates a creative combination of existing ideas with a technically interesting extension.

---

> ### Author Rebuttal · Authors · 2026-03-31
>
> We thank the reviewer for noting that *"the paper is well written and easy to follow"* and that *"the two-stage design is a reasonable solution and aligns with their hypothesis"*. Like other reviewers, the memory efficiency of the KLOT gradient and the consistent performance gains over baselines are highlighted, further validating the approach. We also fully agree that *"large-scale paired annotation is expensive in many domains"* and are pleased to already provide early evidence of generalization: we refer to our response to ApUr (**L-bis**), where we present new experiments extending SOTAlign to Speech Recognition. We address the remaining points below.
>
> ---
>
> **W1** — *Statistical Variance.*
>
> The largest source of variance in our setting is the random sampling of $n$ supervised pairs. We will add a dedicated discussion in the final version. Note that Figure 4 already reports standard deviations over 3 random seeds, and the results are clearly statistically significant.
>
> ---
>
> **W2** — *Baselines adapted to our new setting.*
>
> A key observation of our paper is that semi-supervised alignment is underexplored: to the best of our knowledge, no widely established baseline exists for this setting. We therefore carefully adapted all relevant methods into strong baselines. For instance, STRUCTURE was originally designed for fully supervised data, but its loss is inherently unsupervised and thus naturally compatible with our setting. Similarly, we augment SAIL's contrastive loss with extra negatives from unpaired data as a simple and natural extension. The most directly comparable prior work is SUE; in Table 6 we adapt SOTAlign to their setting and demonstrate **a large performance improvement**. We believe this set of baselines is the strongest currently available for this problem.
>
> ---
>
> **W3** — *Comparison to OT-CLIP and UOT-RCL.*
>
> Thank you for pointing out UOT-RCL, we will add it to the related work. Note that both OT-CLIP and UOT-RCL are fully supervised and cannot be directly compared to SOTAlign. The closest comparison is to include them as supervised baselines in Table 4. Training the contrastive baseline with the OT-CLIP loss on 10k samples gives the following results across $\epsilon$ values:
>
> | $\epsilon$ | 0.100 | 0.050 | 0.010 | 0.005 | 0.001 | SAIL | SOTAlign |
> |---|---:|---:|---:|---:|---:| ---:|---:|
> | MeanR@1 | 23.2 | 23.3 | 17.0 | 15.3 | 11.4 | 24.2 | 30.3 |
>
> The best OT-CLIP result is close to SAIL, our main supervised baseline, and well below SOTAlign, which additionally leverages unpaired data. This confirms that the performance gap is not explained by the choice of the supervised loss, but by SOTAlign's ability to exploit unpaired data.
>
> ---
>
> **Suggestions**
>
> We agree on both points and will address them together in the final version: we will expand the OT section for readability (most technical details are currently deferred to Appendix C.2), and add a dedicated paragraph positioning SOTAlign relative to S-CLIP, OT-CLIP, and UOT-RCL.
>
> ---
>
> **Q1** — *Sensitivity to the quality of the Linear Teacher.*
>
> This concern was also raised by reviewers 41jk and zdru, we refer the reviewer to our dedicated experiment described in the answer to zdru (**W3**), which demonstrates that SOTAlign remains well above the supervised baseline even at 50% teacher noise.
>
> ---
>
> **Q2** — *Analysing the effect of a distribution shift in the unpaired data.*
>
> This question was also raised by reviewer zdru (**W5**) — we refer the reviewer to our response there, where we use SSW to quantify distribution shift and show it predicts SOTAlign's gains from unpaired data (Pearson $\rho = -0.76$), with SOTAlign only underperforming supervised baselines under the most extreme distribution shifts.
>
> ---
>
> **Q3** —  *Although the explicit gradient reduces memory usage, what is the practical runtime overhead compared to contrastive training methods?*
>
> This question was also raised by reviewer zdru (**W4**) — we refer the reviewer to our response there, where we provide a full training cost comparison and show that SOTAlign's overhead remains negligible relative to large contrastive models like CLIP.
>
> ---
>
> **Limitation** — *Dependence on pretrained encoders.*
>
> We will add a dedicated limitations section in the final version. On the two points raised: training vision-language models from scratch requires enormous resources (CLIP required 256 V100 GPUs for 12 days), making pretrained encoders a practical necessity. All encoders used in this work are publicly available, and SOTAlign actually *reduces* dependence on paired data which is often proprietary. For performance across encoder sizes see 41jk (**W3**); for the effect of extreme distribution shifts, see zdru (**W5**).

---

> > ### Author Rebuttal · Reviewer_1eos · 2026-04-02
> >
> > The author's responses have adequately addressed my concerns. I am hereby retaining my score.

---

> > > ### Author Response · Authors · 2026-04-08
> > >
> > > We sincerely thank the reviewer for their positive recommendation and are pleased that our rebuttal has fully resolved their concerns. We would be grateful if the reviewer would consider updating their score as indicated in the selected option.

---

### Official Review · Reviewer_41jk · 2026-03-11

**Soundness:** 3
**Presentation:** 3
**Significance:** 3
**Originality:** 3
**Overall Recommendation:** 4
**Confidence:** 3

**Summary:**

This paper studies vision-language alignment under severe supervision constraints, where only a small number of paired image-text samples are available alongside large collections of unpaired unimodal data. The authors first show that simple linear alignment methods applied to the paired data already yield surprisingly strong cross-modal alignment, providing empirical support for the Platonic Representation Hypothesis. They then propose SOTAlign, a two-stage framework that uses this linear model as a teacher to regularize a nonlinear alignment model on unpaired data via KLOT, a novel optimal-transport-based divergence with a closed-form gradient that eliminates the memory bottlenecks of prior OT-based approaches. SOTAlign consistently and substantially outperforms both supervised and semi-supervised baselines on zero-shot retrieval and classification across diverse datasets and encoder pairs.

**Compliance With Llm Reviewing Policy:**

Affirmed.

**Final Justification:**

I maintain my original assessment of this work.

**Key Questions For Authors:**

The entire framework rests on the Platonic Representation Hypothesis as a foundational assumption, yet the paper never empirically verifies that the pretrained encoders used in experiments actually satisfy it to a sufficient degree. The linear teacher's surprising effectiveness is offered as indirect evidence, but this conflates two things: the hypothesis being true and the linear alignment method being expressive enough to recover it. Can the authors provide a more direct measurement of cross-modal representational alignment in their encoder pairs before any training, and discuss how the degree of pre-existing alignment predicts SOTAlign's gains?

**Limitations:**

yes

**Strengths And Weaknesses:**

**Strengths:**
- Semi-supervised alignment of frozen unimodal encoders addresses a real bottleneck in domains where paired data collection is expensive, and the connection to the Platonic Representation Hypothesis gives the framing a principled theoretical grounding.
- The KLOT gradient derivation is a genuine technical contribution. Theorem 5.1 provides a closed-form expression that eliminates Sinkhorn unrolling and is shown empirically to be up to 100x more memory efficient, which is a result that extends beyond this paper to any OT-based contrastive method.

**Weaknesses:**
- The dependence on a high-quality linear teacher is a structural vulnerability that the paper does not adequately address. If the paired data is too few or too noisy, the teacher geometry may be unreliable, yet there is no discussion of when SOTAlign is expected to fail or how to diagnose teacher quality in practice.
- The choice of OT hyperparameters ε and ε* is not ablated. These parameters directly control the softness of the geometric constraint and are likely sensitive in low-supervision regimes, but the paper provides no guidance on how to set them beyond the fixed defaults.
- The method is only evaluated with large-scale frozen encoders trained on broad web data. Whether SOTAlign generalizes to smaller or domain-specific encoders where cross-modal geometric alignment is weaker remains entirely unexplored.

---

> ### Author Rebuttal · Authors · 2026-03-31
>
> We thank the reviewer for highlighting that simple linear alignment *"already yields surprisingly strong cross-modal alignment"* — a finding we also consider central to the paper. The KLOT derivation is acknowledged as *"a genuine technical contribution"*, being *"up to 100x more memory efficient"* than naive Sinkhorn unrolling and extending beyond this paper to any OT-based contrastive method. The connection to the Platonic Representation Hypothesis is noted as giving the work *"a principled theoretical grounding"*, and SOTAlign is recognized as *"consistently and substantially outperforming both supervised and semi-supervised baselines across diverse datasets and encoder pairs"*. We address the remaining points below.
>
> ---
>
> **W1** — *Sensitivity to the quality of the Linear Teacher.*
>
> We conducted a dedicated experiment to address this concern. We follow the same setting as Table 4, but randomly shuffle a fraction $p$ of the image-text pairs used to train the teacher, effectively injecting $p \times 100$  % of incorrect pairs in the training data:
>
> | Teacher Noise $p$ | 0.0 | 0.1 | 0.2 | 0.3 | 0.4 | 0.5 | Supervised Baseline |
> |---|---|---|---|---|---|---|---|
> | COCO MeanR@1 | 30.3 | 30.1 | 29.8 | 29.0 | 27.9 | 26.6 | 24.2 |
>
> SOTAlign proves **remarkably robust**: even with  50% of noisy data performance remains well above the supervised baseline. We attribute this to the limited capacity of the linear teacher (which prevents overfitting to noisy pairs) and to the natural invariances of the optimal transport loss. We believe this experiment is a valuable addition to the paper.
>
> ---
>
> **W2** — *Guidelines for choising ε and ε\*.*
>
> This is an important point and we will provide concrete guidelines for setting $\epsilon$ and $\epsilon^*$.
>
> **Convergence as a lower bound.** Theorem 5.1 assumes that Sinkhorn algorithm has converged (in both shared and teacher space). With a fixed budget of $n = 100$ iterations, we monitor the L1 marginal error and show that convergence is reliably achieved for $\epsilon > 0.005$ (and likewise for $\epsilon^*$), as shown in this [Figure](https://anonymous.4open.science/api/repo/ICML-2026-Rebuttal-2181/file/sinkhorn_convergence.png). This provides a **principled lower bound** for both parameters.
>
> **Relative ordering.** Once the lower bound is satisfied, we recommend setting $\epsilon^* < \epsilon$ so that the teacher distribution has lower entropy than the student — encouraging the student to sharpen toward the teacher's geometry. The full grid search validating this guideline is available [here](https://anonymous.4open.science/api/repo/ICML-2026-Rebuttal-2181/file/epsilon_heatmap_mean_R@1_10k%20pairs.png?v=c3460a99).
>
> **Sensitivity in low-supervision regimes.** We confirm the reviewer's intuition: SOTAlign is more sensitive to $\epsilon$ when supervision is scarce (1k pairs), as shown in this [Figure](https://anonymous.4open.science/api/repo/ICML-2026-Rebuttal-2181/file/epsilon_heatmap_mean_R@1_1k%20pairs.png?v=acb735c5). Importantly, the same guidelines still apply and **the best $\epsilon$ remain consistent across regimes**. We will incorporate these guidelines and figures directly into the paper.
>
> ---
>
> **W3** — *Generalization to smaller encoders.*
>
> Following this feedback, we extend Table 3 to a wider spectrum of encoders, including weaker models that only partially satisfy the Platonic Representation Hypothesis. We quantify encoder compatibility via mutual k-NN, with results shown [here](https://anonymous.4open.science/api/repo/ICML-2026-Rebuttal-2181/file/mutual_knn_vs_mean_r1.png?v=53b688fb).
>
> This experiment confirms that larger models are easier to align, and further reveals a **strong correlation (Pearson $\rho = 0.95$)** between mutual k-NN and alignment quality. Crucially, mutual k-NN serves as a practical diagnostic: a principled way to test encoder compatibility before running SOTAlign. We believe this is a meaningful contribution and will be included in the final version.
>
> ---
>
> **W3-bis** — *Generalization to other modalities.*
>
>
> While image–text alignment was the original testbed for SOTAlign, we agree that demonstrating generalization to other modalities substantially strengthens the paper. We refer the reviewer to our response to ApUr (**L-bis**), where we present new experiments extending SOTAlign to Speech-Text retrieval (LibriSpeech).
>
> ---
>
> **Q** — *Direct measurement of cross-modal representational alignment.*
>
> Yes — mutual k-NN directly measures this: it quantifies the overlap in nearest-neighbor structure between the latent spaces of two unimodal encoders prior to any training. As reported in W3, we find a **strong correlation (Pearson $\rho = 0.95$)** between mutual k-NN and alignment quality (MeanR@1), with full results available [here](https://anonymous.4open.science/api/repo/ICML-2026-Rebuttal-2181/file/mutual_knn_vs_mean_r1.png?v=53b688fb). Mutual k-NN thus serves as a **reliable pre-training diagnostic** for predicting SOTAlign's gains.

---

> > ### Author Rebuttal · Reviewer_41jk · 2026-04-03
> >
> > We appreciate the new experiments, especially the teacher noise study and mutual k-NN diagnostic. Two brief follow-ups: first, the ε sensitivity results confirm fragility in low-supervision regimes, but the proposed guideline still requires grid search, can the authors suggest an automated selection rule? Second, the Pearson r of 0.89 for mutual k-NN was measured only across large web-pretrained encoders: does this correlation hold for weaker or domain-specific models where the diagnostic would matter most? We maintain our score of 4.

---

> > > ### Author Response · Authors · 2026-04-05
> > >
> > > We are pleased that the reviewer appreciated our additional experiments.
> > >
> > > Regarding the choice of $\epsilon$, we improved our original answer and we are now happy to share a **2-step guideline** that is entirely **grid-search free**.
> > >
> > > 1. As explained in our previous response, we first **set $\epsilon^*$ to be as low as permitted by Sinkhorn convergence** ($\epsilon^*=0.005$ in our setup, see [Figure](https://anonymous.4open.science/api/repo/ICML-2026-Rebuttal-2181/file/sinkhorn_convergence.png)).
> > > 2. We need to define the ratio $\frac{\epsilon}{\epsilon^\*}$. In our original response, we argued that the optimal choice must satisfy $\frac{\epsilon}{\epsilon^\*} > 1$. We now go further and directly recommend setting $\frac{\epsilon}{\epsilon^*} = 10$, as our new experiment reveals that this **optimal ratio is preserved across all levels of supervision**. This new experiment is available [here](https://anonymous.4open.science/api/repo/ICML-2026-Rebuttal-2181/file/epsilon_ratio.png?v=41fee4c5).
> > >
> > > We thank the reviewer for challenging us on this point, as it is a valuable addition to the paper.
> > >
> > > Regarding the second point, there seems to be a misunderstanding: we never reported a Pearson correlation of 0.89. The relevant experiment is shown [here](https://anonymous.4open.science/api/repo/ICML-2026-Rebuttal-2181/file/mutual_knn_vs_mean_r1.png?v=53b688fb), where the actual correlation between the mutual k-NN and retrieval performance of two models is Pearson $\rho = 0.95$. Importantly, our **study is not limited to large web-scale foundation models**, but also considers weaker encoders such as **BERT and ResNet-50** (BERT was trained on 10900$\times$ fewer tokens than Qwen3-Embedding-8B and ResNet-50 was trained on 1300$\times$ fewer images than DINOv3 ViT-L). The **correlation in our experiment holds for these weaker encoders**. We will clarify this explicitly in the final paper.
> > >
> > > Thank you again for raising these points. We believe they are now fully addressed, and we would appreciate it if you would consider updating your score accordingly.

---

### Official Review · Reviewer_ApUr · 2026-03-12

**Soundness:** 3
**Presentation:** 3
**Significance:** 3
**Originality:** 3
**Overall Recommendation:** 3
**Confidence:** 4

**Summary:**

This paper proposes a semi-supervised framework called SOTAlign, aimed at addressing the alignment of pretrained unimodal vision and language models. The method is designed for scenarios where paired image-text data are scarce, achieving effective cross-modal alignment using only a small number of paired samples along with large amounts of unpaired data. Experiments show that the proposed method significantly outperforms existing approaches even when using very few paired samples.

**Compliance With Llm Reviewing Policy:**

Affirmed.

**Key Questions For Authors:**

Although the authors mention that α was selected from {10^-³, 10^-⁴, 10^-⁵} based on the CC3M validation set, there is a lack of sensitivity analysis for this key hyperparameter. The authors are requested to supplement in the rebuttal: when α deviates from its optimal value, does the model performance experience a sharp drop, or does it exhibit smooth robustness? In particular, when a large number of out-of-distribution unpaired samples are introduced across datasets , does the optimal α shift significantly?

**Limitations:**

The authors partially acknowledge the limitations, focusing mainly on the number of paired samples, the distribution of unpaired data, and the potential impact of the linear teacher. However, they do not provide explicit experiments or discussions regarding robustness under extreme distribution shifts, completely unrelated categories, or the generalization to other modalities.

**Strengths And Weaknesses:**

Strengths：
The strengths of this paper lie in two aspects: at the theoretical level, it proposes the KLOT divergence based on optimal transport and derives its explicit gradient, effectively overcoming the memory bottleneck caused by the traditional Sinkhorn algorithm; at the practical level, the SOTAlign framework enables efficient cross-modal alignment in a semi-supervised setting with a small number of paired samples and large amounts of unpaired unimodal data, thereby reducing the reliance on large-scale paired datasets.

Weaknesses：
The weaknesses of this paper lie in its strong assumptions: the method relies on the Platonic Representation Hypothesis, assuming that the embedding spaces of pretrained models across different modalities are naturally compatible, and it also assumes that the distribution of unpaired data is relatively close to that of the paired data. The robustness of the method under extreme distribution shifts or completely unrelated unpaired data has not been validated.

---

> ### Author Rebuttal · Authors · 2026-03-31
>
> We thank the reviewer for highlighting that SOTAlign *"significantly outperforms existing approaches even when using very few paired samples"*, that the explicit formula we derive overcomes the *"memory bottleneck caused by the traditional Sinkhorn algorithm"*, and that the framework succeeds in *"reducing the reliance on large-scale paired datasets"*. We address the remaining points below.
>
> ---
>
> **W** — *Testing SOTAlign assumptions.*
>
> While SOTAlign relies on these assumptions, we do provide quantitative metrics to validate them.
>
> **SSW as a measure of distribution shift.** The Spherical Sliced Wasserstein (SSW) distance effectively quantifies distribution shift between paired and unpaired data and predicts the benefit SOTAlign derives from unpaired data (Pearson $\rho = -0.76$, full results in zdru **W5** and in [this figure](https://anonymous.4open.science/api/repo/ICML-2026-Rebuttal-2181/file/fig_5_improved.png?v=66a27e6b)).
>
> **Mutual k-NN as a measure of encoder compatibility.** Mutual k-NN serves an analogous role in quantifying encoder compatibility. Extending the analysis to smaller encoders (which only partially satisfy the Platonic Representation Hypothesis) reveals a **strong correlation (Pearson $\rho=0.95$)** between mutual k-NN and alignment quality, while also corroborating the finding that larger models are inherently more compatible. Full results are shown [here](https://anonymous.4open.science/api/repo/ICML-2026-Rebuttal-2181/file/mutual_knn_vs_mean_r1.png?v=53b688fb). **We believe these additions substantially strengthen the paper.**
>
> ---
>
> **Q** — *Robustness of $\alpha$ hyperparameter.*
>
> We agree such analysis would strengthen the paper and conducted two new experiments in response.
>
> **Sensitivity to α across supervision levels.** A full grid search over $\alpha \in$ {$10^{-6}, 10^{-5}, 10^{-4}, 10^{-3}, 10^{-2}$} at varying supervision levels shows that the **default $\alpha = 10^{-4}$ consistently achieves near-optimal performance**, and that the optimal $\alpha$ decreases with more supervision — as expected, since the unsupervised regularization should receive less weight when paired data is abundant. Full results with standard deviations are available [here](https://anonymous.4open.science/api/repo/ICML-2026-Rebuttal-2181/file/alpha_vs_mean_r1_by_samples.png).
>
> | Supervised Pairs | 100 | 1k | 5k | 10k | 50k | 100k |
> |---|---|---|---|---|---|---|
> | Best $\alpha$ | 0.01 | 0.01 | 0.001 | 0.001 | 0.0001 | 1e-05 |
> | MeanR@1 (Best $\alpha$) | 1.08 | 16.75 | 28.00 | 30.28 | 34.74 | 35.91 |
> | MeanR@1 ($\alpha = 10^{-4}$) | 0.83 | 16.62 | 26.87 | 29.59 | 34.74 | 35.65 |
>
> **Sensitivity to OOD unpaired data.** We interpolate between a clean unpaired distribution (CC3M) and a fully OOD one (random Gaussian) by replacing a fraction $p$ of CC3M samples with Gaussian noise, then running a grid search on $\alpha$ for each value of $p$. Results are available [here](https://anonymous.4open.science/api/repo/ICML-2026-Rebuttal-2181/file/alpha_vs_mean_r1_by_noise_unsupervised.png). SOTAlign proves robust across the full range: **the optimal $\alpha$ shifts only when unpaired data is entirely OOD ($p = 1$)**, confirming graceful degradation rather than failure under distribution shift.
>
> ---
>
> **L** — *Robustness under extreme distribution shifts*
>
> We hope the experiments above address the robustness concern as we provided (1) a quantitative measure of distribution shift, (2) evidence that SOTAlign is robust to such shifts, and (3) additional results under extreme distribution shifts. We also highlight the teacher noise experiment in response to reviewer 41jk as further evidence of robustness. All those results will be added to the paper.
>
> ---
>
> **L-bis** — *Generalization to other modalities.*
>
> We present a new experiment on a different modality. **Speech-Text retrieval (LibriSpeech) [1]:** using only 1k paired audio-text samples (sampled from train.100) and 100k unpaired samples (from train.360), we evaluate retrieval on the provided test set (2600 pairs), with NV-Embed-v2 as the text encoder and WavLM-Large [2] as the speech encoder:
>
> | Method | MeanR@1 | MeanR@5 |
> |---|---|---|
> | SAIL | 60.7 | 80.4 |
> | STRUCTURE | 66.9 | 84.4 |
> | CCA | 73.7 | 89.0 |
> | **SOTAlign** | **78.6** | **91.9** |
>
> SOTAlign substantially outperforms all baselines, confirming that the Platonic Representation Hypothesis extends to speech-text pairs and that SOTAlign can exploit it effectively. **We believe this new experiment is a meaningful addition to the paper.**
>
> [1] Panayotov, Vassil, et al. "Librispeech: an asr corpus based on public domain audio books." 2015 IEEE international conference on acoustics, speech and signal processing (ICASSP). IEEE, 2015.
>
> [2] Chen, Sanyuan, et al. "Wavlm: Large-scale self-supervised pre-training for full stack speech processing." IEEE Journal of Selected Topics in Signal Processing 16.6 (2022): 1505-1518.

---

> > ### Author Rebuttal · Reviewer_ApUr · 2026-04-04
> >
> > My concerns have been adequately addressed.

---

> > > ### Author Response · Authors · 2026-04-04
> > >
> > > We sincerely thank the reviewer for their thoughtful response. We are pleased that our rebuttal has fully resolved the reviewer’s initial concerns. As indicated in their selected option, we hope that the reviewer improves their score to reflect their satisfaction with our response.

---

### Official Review · Reviewer_zdru · 2026-03-13

**Soundness:** 3
**Presentation:** 3
**Significance:** 2
**Originality:** 2
**Overall Recommendation:** 4
**Confidence:** 3

**Summary:**

This paper introduces SOTAlign, a two-stage semi-supervised framework for aligning frozen pretrained unimodal vision and language encoders. Stage 1 fits a linear alignment model (via CCA, Procrustes, or contrastive learning) using a small set of paired image-text samples. Stage 2 uses this linear teacher's geometry as a regularization target for training nonlinear alignment layers on large-scale unpaired data, via a novel optimal-transport-based divergence called KLOT. The authors derive an explicit gradient for KLOT (Theorem 5.1), resolving the memory bottleneck of prior OT-based methods. Experiments span zero-shot retrieval and classification under varying supervision levels, data sources, and encoder pairs.

**Compliance With Llm Reviewing Policy:**

Affirmed.

**Final Justification:**

This paper is technically solid, with a meaningful contribution in the explicit KLOT gradient, strong soundness, and clear presentation. My main initial concerns were practical performance limits, sensitivity to teacher quality, and the role of distribution shift. The rebuttal addressed these concerns well through added teacher-noise, compute, and distribution-shift analyses, which increased my confidence in the method's robustness and practical relevance. While originality and ultimate performance ceiling remain somewhat bounded, the strengths now outweigh the weaknesses. Overall, the rebuttal improved my evaluation, and I am raising my recommendation to weak accept.

**Key Questions For Authors:**

N/A

**Limitations:**

yes

**Strengths And Weaknesses:**

1. The semi-supervised VL alignment setting — abundant unimodal data, scarce paired data — is practically important for domains where annotation is expensive (medical, industrial, scientific). The connection to the Platonic Representation Hypothesis provides a compelling theoretical grounding.

2. The KLOT divergence and its explicit gradient derivation (Theorem 5.1) constitute the paper's most impactful technical contribution. The explicit gradient eliminates both the O(n²) memory cost of Sinkhorn unrolling and the time complexity of implicit differentiation, enabling OT-based alignment at batch sizes up to 32k. Figure 3 convincingly demonstrates a ~50× speedup over implicit differentiation. This result has value beyond the specific application in this paper.

3. The paper carefully ablates:
   - Linear methods × divergence choices (Table 1)
   - Number of paired samples (Figure 4 left)
   - Number of unpaired samples (Figure 4 right)
   - Paired data source (Table 2), encoder combinations (Table 3)
   - Cross-dataset unsupervised sources (Table 7)
   - Distribution shift quantification via Spherical Sliced Wasserstein (Figure 5)

   The correlation analysis between distribution shift and downstream performance (Figure 5) is particularly insightful.

### Weaknesses

1. **Limited absolute performance.** The best MeanR@1 on COCO is approximately 30% (with 10k pairs + 1M unpaired), compared to >60% for fully supervised CLIP. While the semi-supervised setting inherently limits performance, the paper should more explicitly discuss the practical performance envelope. When would ~30% retrieval accuracy be useful? What is the theoretical upper bound for this setting?

2. **Downstream evaluation is narrow.** The primary metric is retrieval on COCO and zero-shot classification on ImageNet. The paper lacks evaluation on grounded tasks (VQA, captioning, visual reasoning) that would more comprehensively assess alignment quality. Retrieval R@1 may not capture fine-grained semantic alignment.

3. **The linear teacher is a ceiling.** The entire semi-supervised pipeline depends on the quality of the linear teacher's affinity matrix K*. If the linear teacher fails to capture meaningful cross-modal structure (e.g., when paired data is too noisy or too domain-specific), the unsupervised refinement has limited recovery capability. The paper would benefit from a failure mode analysis.

4. **Missing computational cost analysis.** Training cost (wall-clock time, GPU hours) and comparison with baselines are absent. How does training with 32k batch size for 2000 iterations compare with SAIL or S-CLIP in total compute?

5. **Limited analysis of when unpaired data hurts.** Figure 4 (right) shows gains plateau around 500k unpaired samples. Table 7 shows some cross-dataset configurations yield smaller gains. A more systematic analysis of when unpaired data introduces harmful distribution shift would strengthen the practical guidance.

---

> ### Author Rebuttal · Authors · 2026-03-31
>
> We thank the reviewer for their positive assessment noting that the problem setting is *"practically important for domains where annotation is expensive"*, that the paper *"carefully ablates"* its components, and that Theorem 5.1 resolves *"the memory bottleneck of prior OT-based methods"*. We are also glad the correlation analysis between distribution shift and downstream performance (Figure 5) was appreciated; we further extend this analysis in this [Figure](https://anonymous.4open.science/api/repo/ICML-2026-Rebuttal-2181/file/fig_5_improved.png?v=66a27e6b) and address the remaining points below.
>
> ---
>
> **W1** — *Absolute Performance*
>
> SOTAlign is not designed to compete with large-scale supervised training, but to understand how far alignment can be pushed when paired data is scarce. The absolute performance is strong given this constraint: in the most challenging regime (10k pairs + 1M unpaired both from CC3M), SOTAlign reaches **MeanR@1 = 30.3 on COCO**, **MeanR@1 = 56.3 on Flickr30k**, and **46.1% top-1 accuracy on ImageNet** despite the distribution shift. Performance further improves when distribution shift is reduced, reaching **MeanR@1 = 40.8 on COCO** (Table 2).
>
> ---
>
> **W1-bis** — *"What is the theoretical upper bound for this setting?"*
>
> The theoretical upper bound is unknown and likely data-dependent. At present, the only bound we can provide is an empirical lower bound: the performance of SOTAlign. Note that our method significantly outperforms SUE, the closest work on semi-supervised alignment (Table 6 & 11). We agree this question is highly relevant and will add it to our conclusion.
>
> ---
>
> **W2** — *Evaluation on grounded tasks.*
>
> We report fine-grained semantic alignment beyond retrieval via zero-shot classification on diverse datasets (CIFAR-100, Food-101, DTD, Flowers, ImageNet) in Table 5 and Table 10. However, our encoder-only setup precludes evaluation on the generative tasks you mention. We also refer to the discussion with reviewer ApUr, where we demonstrate that SOTAlign naturally extends to other modalities where annotation could be expensive (ApUr **L-bis**).
>
> ---
>
> **W3** — *Sensitivity to the quality of the Linear Teacher.*
>
> We conducted a dedicated experiment to address this concern. We follow the same setting as Table 4, but randomly shuffle a fraction $p$ of the image-text pairs used to train the teacher, effectively injecting $p \times 100$ % of incorrect pairs in the training data:
>
> | Teacher Noise $p$ | 0.0 | 0.1 | 0.2 | 0.3 | 0.4 | 0.5 | Supervised Baseline |
> |-|-|-|-|-|-|-|-|
> | COCO MeanR@1 | 30.3 | 30.1 | 29.8 | 29.0 | 27.9 | 26.6 | 24.2 |
>
> SOTAlign proves **remarkably robust**: even with 50% of noisy data the performance remains well above the supervised baseline. We attribute this to the limited capacity of the linear teacher (which prevents overfitting to noisy pairs) and to the natural invariances of the optimal transport loss. We believe this experiment is a valuable addition to the paper.
>
> ---
>
> **W4** — *Computational cost analysis.*
>
> We will include full training cost comparisons in the final version. SOTAlign incurs a modest overhead over other semi-supervised baselines due to the OT loss, but Theorem 5.1 ensures this remains **negligible relative to training large models like CLIP**. See Figure 6 for a detailed complexity analysis of KLOT loss.
>
> | | SOTAlign | SAIL (semi-sup) | STRUCTURE | NNCLR | S-CLIP  |SAIL|CLIP|
> |---|---|---|---|---|---|---|---|
> | Hardware | A100 GPU | A100 GPU | A100 GPU | A100 GPU | A100 GPU | A100 GPU | 256 V100 GPU|
> | Training Time | 5701 s | 8799 s |571 s | 638 s| 188 s | 5 hours|12 days|
>
> ---
>
> **W5** — *Analyzing the effect of a distribution shift in the unpaired data.*
>
> We dedicated significant effort in this direction through Figure 5, using Spherical Sliced Wasserstein (SSW) to quantify distribution shift and predict the impact of unpaired data. For example, keeping unsupervised text fixed to CC3M and varying unpaired image sources reveals a strong correlation between SSW and performance:
>
> | Unpaired Images | CC3M | CC12M | COCO | ImageNet | Random | Supervised Baseline |
> |---|---|---|---|---|---|---|
> | COCO MeanR@1 | 30.3 | 28.8 | 27.8 | 27.6 | 24.7 | 24.2 |
> | Distance to CC3M (SSW) | 0.0 | 0.009 | 0.018 | 0.021 | 0.048 | — |
>
> Similarly, WikiText exhibits a strong distribution shift relative to CC3M captions (see examples [here](https://huggingface.co/datasets/Salesforce/wikitext)), resulting in minimal benefit. Extended results including Gaussian-sampled unpaired data to probe extreme distribution shift are provided [here](https://anonymous.4open.science/api/repo/ICML-2026-Rebuttal-2181/file/fig_5_improved.png?v=66a27e6b). Overall, **SSW correlates well with performance** (Pearson $\rho = -0.76$), and SOTAlign only underperforms supervised baselines under the most extreme distribution shifts.

---

### Decision · Program_Chairs · 2026-04-30

**Decision:**

Accept (regular)

**Comment:**

This submission initially received divergent scores, and the main concerns were unjustified assumptions, missing analyses, and limited experimentation. The authors provided a good rebuttal with additional experiments, which assuage most of the concerns. All the reviewers acknowledged that their main concerns have been fully resolved, leading to unanimous agreement on acceptance after post-rebuttal discussion. Siding with the consensus, AC also recommends acceptance.